# LuxT controls specific quorum-sensing-regulated behaviors in *Vibrionaceae* spp. via repression of *qrr*1, encoding a small regulatory RNA

**Michaela J. Eickhoff**[1], **Chenyi Fei**[1,2], **Xiuliang Huang**[1,3], **Bonnie L. Bassler**[1,3]*

**1** Department of Molecular Biology, Princeton University, Princeton, New Jersey, United States of America, **2** Lewis-Sigler Institute for Integrative Genomics, Princeton University, Princeton, New Jersey, United States of America, **3** Howard Hughes Medical Institute, Chevy Chase, Maryland, United States of America

* bbassler@princeton.edu

**Data Availability Statement:** All relevant data are within the manuscript and its Supporting Information files.

## Abstract

Quorum sensing (QS) is a process of chemical communication bacteria use to transition between individual and collective behaviors. QS depends on the production, release, and synchronous response to signaling molecules called autoinducers (AIs). The marine bacterium *Vibrio harveyi* monitors AIs using a signal transduction pathway that relies on five small regulatory RNAs (called Qrr1-5) that post-transcriptionally control target genes. Curiously, the small RNAs largely function redundantly making it difficult to understand the necessity for five of them. Here, we identify LuxT as a transcriptional repressor of *qrr*1. LuxT does not regulate *qrr*2-5, demonstrating that *qrr* genes can be independently controlled to drive unique downstream QS gene expression patterns. LuxT reinforces its control over the same genes it regulates indirectly via repression of *qrr*1, through a second transcriptional control mechanism. Genes dually regulated by LuxT specify public goods including an aerolysin-type pore-forming toxin. Phylogenetic analyses reveal that LuxT is conserved among *Vibrionaceae* and sequence comparisons predict that LuxT represses *qrr*1 in additional species. The present findings reveal that the QS regulatory RNAs can carry out both shared and unique functions to endow bacteria with plasticity in their output behaviors.

## Author summary

Bacteria communicate and count their cell numbers using a process called quorum sensing (QS). In response to changes in cell density, QS bacteria alternate between acting as individuals and participating in collective behaviors. *Vibrio harveyi* is used as a model organism to understand QS-mediated communication. Five small RNAs lie at the heart of the *V. harveyi* QS system, and they regulate the target genes that underlie the QS response. The small RNAs largely function redundantly making it difficult to understand why *V. harveyi* requires five of them. Here, we discover a regulator, called LuxT, that exclusively represses the gene encoding one of the QS small RNAs. LuxT regulation of one QS small

**Funding:** This work was supported by the Howard Hughes Medical Institute, National Institutes of Health (NIH) Grant 5R37GM065859, and National Science Foundation Grant MCB-1713731 (to BLB). MJE was supported by NIH graduate training grant NIGMS T32GM007388. The funders had no role in study design, data collection and analysis, decision to publish, or preparation of the manuscript.

**Competing interests:** The authors have declared that no competing interests exist.

RNA enables unique control of a specific subset of QS target genes. LuxT is broadly conserved among *Vibrionaceae*. Our findings show how redundant regulatory components can possess both common and unique roles that provide bacteria with plasticity in their behaviors.

## Introduction

Bacteria can coordinate gene expression on a population-wide scale using a process of cell-cell communication called quorum sensing (QS). QS depends on the production, release, and detection of signal molecules called autoinducers (AIs). Because AIs are self-produced by the bacteria, as cell density increases, extracellular AI levels likewise increase. Bacteria respond to accumulated AIs by collectively altering gene expression, and in turn, behavior. QS-regulated processes include bioluminescence, biofilm formation, and the secretion of virulence factors [1, 2].

*Vibrio harveyi* is a model marine bacterium that uses QS to regulate over 600 genes [3–8]. *V. harveyi* produces and responds to three AIs, which act in parallel. The LuxM synthase produces AI-1 (*N*-(3-hydroxybutanoyl)-L-homoserine), LuxS produces AI-2 ((2*S*,4*S*)-2-methyl-2,3,3,4-tetrahydroxytetrahydrofuran-borate), and CqsA produces CAI-1 ((*Z*)-3-aminoundec-2-en-4-one) [3, 9–16]. The three AIs are recognized by the cognate receptors LuxN, LuxPQ, and CqsS, respectively [3, 4, 13, 14]. At low cell density (LCD, Fig 1A), when little AI is present, the unbound receptors act as kinases that transfer phosphate to the phosphorelay protein LuxU, which shuttles the phosphoryl group to the response regulator, LuxO [4, 6, 17, 18]. LuxO-P, together with the alternative sigma factor $\sigma^{54}$, activates expression of genes encoding five non-coding small regulatory RNAs (sRNAs), Qrr1-5, that function post-transcriptionally [6, 19, 20]. The five Qrr sRNAs promote translation of *aphA* and repress translation of *luxR*, encoding the LCD and high cell density (HCD) QS master transcriptional regulators, respectively (Fig 1A) [19, 21–26]. When the Qrr sRNAs are produced, individual behaviors are undertaken and the luciferase operon (*luxCDABE*), responsible for the canonical bioluminescence QS output in *V. harveyi*, is not expressed. At HCD (Fig 1B), when the AIs bind to their cognate receptors, the receptors' kinase activities are inhibited, allowing their phosphatase activities to dominate. Consequently, phospho-flow through the QS circuit is reversed [27]. Dephosphorylated LuxO is inactive. Thus, Qrr1-5 are not produced, *aphA* translation is not activated, and *luxR* translation is not repressed (Fig 1B). In this state, LuxR is produced, and it controls expression of genes underpinning group behaviors. Notably, LuxR activates expression of *luxCDABE*, causing *V. harveyi* cells to make light at HCD [14].

The five *V. harveyi* Qrr sRNAs have high sequence identity and they are predicted to possess similar secondary structures with four stem loops [19]. Mechanistic studies of Qrr3 as the exemplar Qrr showed it regulates translation of its different target mRNAs by four mechanisms, all mediated by the chaperone Hfq; repression via catalytic degradation of the mRNA target, repression via coupled degradation of Qrr3 with the mRNA target, repression through sequestration of the mRNA target, and activation via revelation of the mRNA ribosome-binding site [25]. In addition to *aphA* and *luxR*, the Qrr sRNAs also feedback to repress *luxO* and *luxMN* translation [28, 29]. Microarray analyses following *qrr* overexpression revealed 16 additional Qrr-controlled target mRNAs [30].

The extreme relatedness of the Qrr sRNAs, coupled with their similar QS-controlled production patterns, has made it difficult to assign any unique role to a particular Qrr sRNA. Nonetheless, among the Qrr sRNAs, Qrr1 stands out: it lacks nine nucleotides in stem loop 1

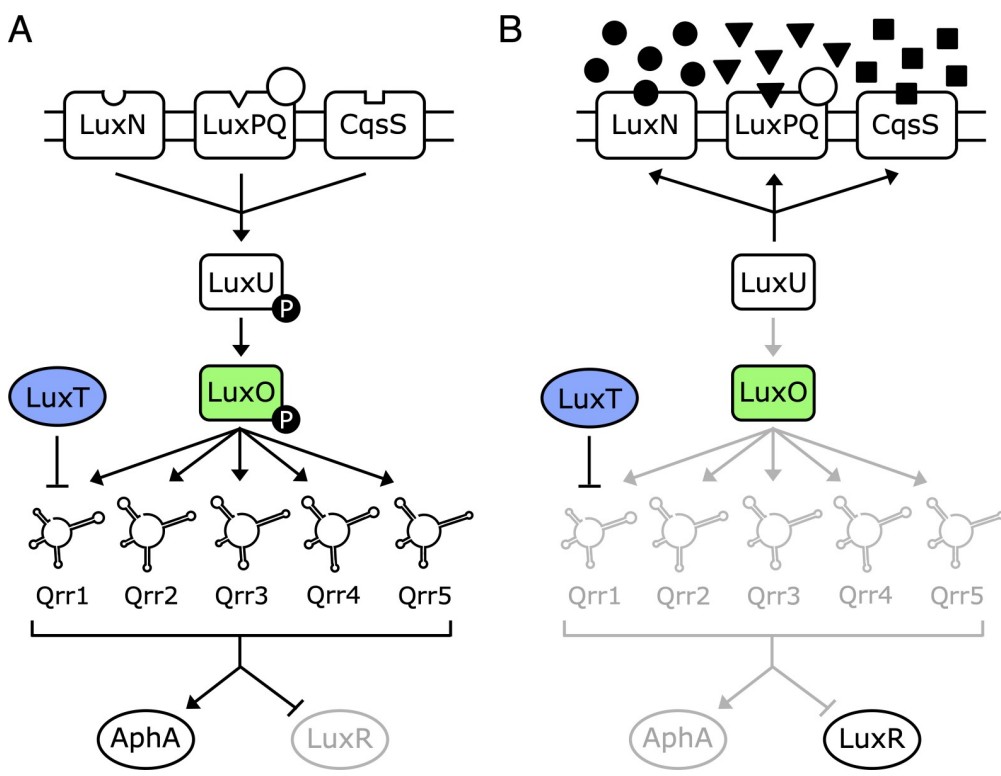

**Fig 1. Model of the *V. harveyi* QS system. (A)** LCD and **(B)** HCD. See text for details.

that are present in Qrr2-5 [19, 26, 30]. Due to this difference, Qrr1 does not regulate *aphA* and two of the other known target mRNAs [26, 30]. Qrr2-5 regulate an identical set of target mRNAs [30]. Thus, the failure of Qrr1 to control one subset of mRNAs is the only functional difference known among the Qrr sRNAs. Also of note is the position of *qrr*1 in the *V. harveyi* genome: *qrr*1 is located immediately upstream of *luxO*, oriented in the opposite direction [19, 20]. No other *qrr* genes reside near known QS genes.

Predicted LuxO-P and σ$^{54}$ binding sites lie upstream of each *qrr* gene. The sites vary in sequence and relative position with respect to the *qrr* transcriptional start sites. Other than these sites, there is little sequence similarity between *qrr* promoter regions [19, 20]. There also exist hallmarks of transcription factor binding sites upstream of *qrr* genes, which differ in every case, hinting that unique factors could regulate each *qrr* gene [19]. Indeed, while all the Qrr sRNAs are made at LCD, they exhibit distinct production profiles. Specifically, in order of highest to lowest expression: Qrr4 > Qrr2 > Qrr3 > Qrr1 > Qrr5 [19]. The strength by which each Qrr sRNA represses *luxR* translation, and therefore downstream bioluminescence emission, correlates with Qrr production level: Qrr4 is the strongest repressor of light production, while Qrr1 and Qrr5 are the weakest [19]. When introduced into *Escherichia coli*, all five *qrr* sRNA genes are activated to high levels by LuxO D61E, a LuxO-P mimetic, suggesting that regulation by additional factors, that are not present in *E. coli*, occurs in *V. harveyi* [19]. Investigating the possibility that other regulators are involved in *qrr* control *in vivo* is the subject of the present work.

LuxT is a 17 kDa transcriptional regulator of the AcrR/TetR family, initially identified as a protein that binds strongly to DNA containing the region upstream of the *V. harveyi luxO* gene [31, 32]. An approximate 50 bp region that is bound by LuxT was discovered [31]. A

follow-up report showed that LuxT activates light production in *V. harveyi*, the presumption being that LuxT functioned via repression of *luxO* [32]. At the time of this earlier study, the Qrr sRNAs had not been discovered and LuxO was assumed to be a repressor of bioluminescence. Thus, the logic of the first LuxT manuscripts were: LuxT represses *luxO*, and LuxO represses luciferase.

Research undertaken since the original LuxT publications has led to the current understanding of mechanisms underlying *V. harveyi* QS-controlled gene regulation (Fig 1). Key is that LuxO phosphorylation, not *luxO* expression, is regulated (Fig 1). This incongruity inspired us to reconsider the earlier findings concerning LuxT. Here, we explore the role of LuxT in *V. harveyi* QS with a focus on its connection to *qrr*1. We show that LuxT does indeed bind upstream of *luxO* at the site originally identified [31]. However, LuxT does not regulate *luxO*. While the experiments in the initial manuscripts were rigorously performed and interpreted appropriately, the authors could not have known that the gene encoding Qrr1 is located adjacent to *luxO*. We discover that the LuxT binding region is located within the *qrr*1 promoter. Indeed, we show that LuxT represses the transcription of *qrr*1 at LCD. LuxT does not repress *qrr*2-5. Relative to wild-type (WT) *V. harveyi*, in a Δ*luxT* mutant, *qrr*1 is expressed more highly at LCD. As a consequence, Qrr1 is available to post-transcriptionally regulate its target genes, including a gene encoding an extracellular protease (*VIBHAR_RS11785*), a gene encoding a pore-forming aerolysin toxin (*VIBHAR_RS11620*), a gene encoding a chitin deacetylase (*VIBHAR_RS16980*), and a gene specifying a component involved in capsular polysaccharide secretion (*VIBHAR_RS25670*) [30]. In addition to indirect activation of these genes via repression of *qrr*1, LuxT also activates transcription of these same four genes. Finally, we show that LuxT repression of *qrr*1 transcription is not specific to *V. harveyi*. LuxT also represses *qrr*1 in *Aliivibrio fischeri*, a species that, interestingly, harbors only a single Qrr sRNA: *qrr*1. Phylogenetic analyses show that *luxT* is conserved among *Vibrionaceae* and suggest that LuxT may repress *qrr*1 in other species within the *Vibrionaceae* family. Together, our results support a new QS model that incorporates LuxT and provides a mechanism for the unique control of one of the Qrr sRNA genes, *qrr*1. This newly revealed regulatory arrangement shows how Qrr1 controls downstream targets distinct from those controlled by the other Qrr sRNAs.

# Results

## LuxT binds upstream of *luxO* but does not repress *luxO* transcription

In the original works that identified and studied *V. harveyi* LuxT, DNA binding assays revealed the LuxT binding site to be a roughly 50 bp region lying 76 bp upstream of the *luxO* start codon [31]. (We note that in those reports, the site was designated to be 117 bp upstream of *luxO*, due to initial mis-annotation of the *luxO* start codon.) By assaying changes in light production, the authors concluded that LuxT represses *luxO* transcription [32]. This result is curious because our subsequent work showed that *luxO* is transcribed constitutively and only its phosphorylation state changes in response to QS signaling [18, 33]. Indeed, all fluctuations in LuxO levels in *V. harveyi* have been ascribed to intrinsic noise [33]. To confirm that LuxT binds upstream of *luxO*, we conducted electrophoretic mobility shift assays (EMSAs) using purified LuxT protein. Analogous to the results described by Lin et al. [31], LuxT caused a shift of a 95 bp DNA probe encompassing the *luxO* promoter region, whereas no significant binding to a control DNA probe occurred (Fig 2A). In the context of the 95 bp *luxO* promoter probe, randomizing the DNA sequence of the identified 50 bp LuxT binding region nearly eliminated LuxT binding (S1 Fig). Also consistent with the initial findings, deletion of *luxT* caused an ~11-fold reduction in light production by *V. harveyi* at LCD, indicating that LuxT is a LCD activator of luciferase (Fig 2B) [32]. At HCD ($OD_{600} > 1$), the WT and Δ*luxT V. harveyi*

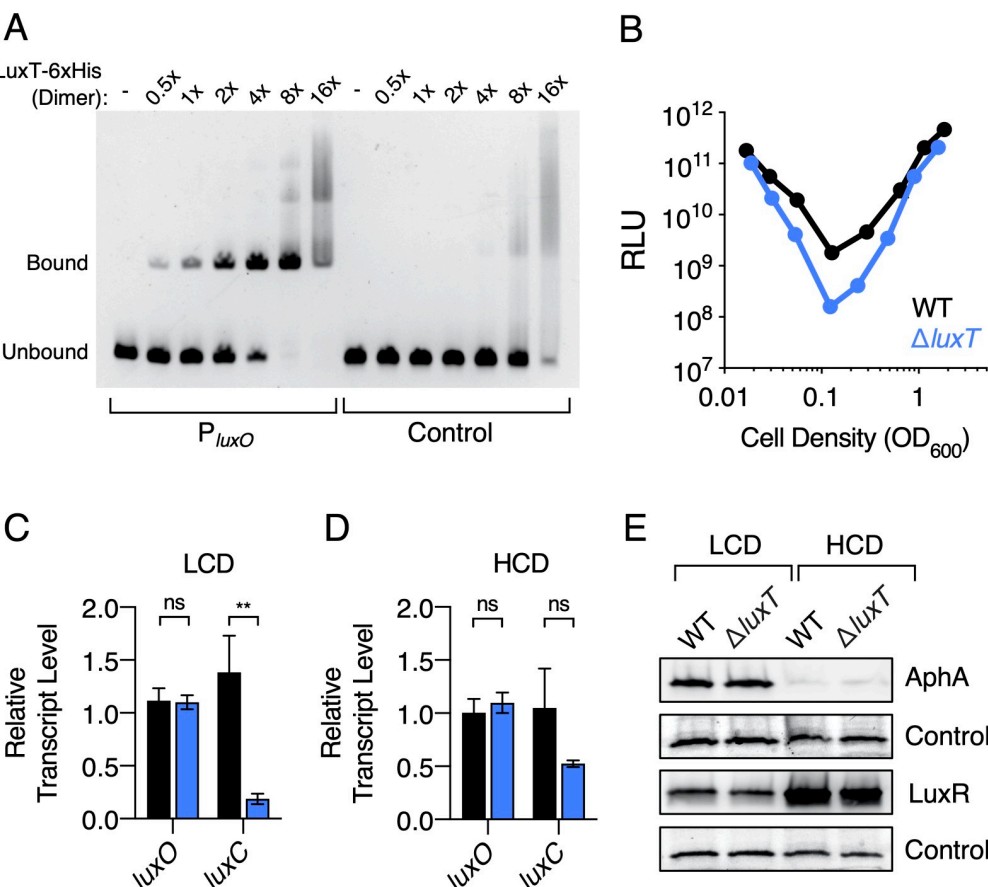

**Fig 2. LuxT binds upstream of *luxO*, but it does not repress *luxO*. (A)** EMSAs showing binding of LuxT-6xHis to 95 bp DNA fragments consisting of the *luxO* promoter (left) or control (*E. coli lacZ*) DNA (right). Reaction mixtures contained 20 nM DNA probe and the indicated relative concentrations of the LuxT-6xHis dimer:- = no protein, 1x = 20 nM, 16x = 320 nM. **(B)** Density-dependent bioluminescence emission from WT (black) or Δ*luxT* (blue) *V. harveyi*. Relative light units (RLU) are counts/min mL$^{-1}$ per OD$_{600}$. Error bars represent standard deviations of the means of *n* = 3 biological replicates. Standard deviations that are smaller than the symbols are not shown. **(C)** qRT-PCR of *luxO* and *luxC* at LCD (OD$_{600}$ = 0.05) of WT (black) and Δ*luxT* (blue) *V. harveyi*. Error bars represent standard deviations of the means of *n* = 3 biological replicates. Unpaired two-tailed *t* tests with Welch's correction were performed comparing WT to Δ*luxT*. *p*-values: ns ≥ 0.05, ** < 0.01. **(D)** As in C at HCD (OD$_{600}$ = 1). **(E)** Western blots of AphA-3xFLAG (top) and 3xFLAG-LuxR (3$^{rd}$ panel from top) in WT and Δ*luxT V. harveyi* at LCD (OD$_{600}$ = 0.01) and HCD (OD$_{600}$ = 1). Total proteins were visualized on a stain-free gel before transfer (2$^{nd}$ and bottom panels), and a dominant band serves as a loading control.

strains exhibited similar light production profiles (Fig 2B). Therefore, LuxT activation of luciferase expression is cell-density dependent, indicating a possible role for QS.

The implication from the above findings, based on the original work, is that LuxT functions via repression of *luxO*. To investigate this possibility, we measured *luxO* transcript levels in WT and Δ*luxT V. harveyi*. We also measured transcript levels of *luxC*, the first gene in the luciferase operon. There were no detectable differences in *luxO* transcript levels in the WT and Δ*luxT* strains at either LCD or HCD (Fig 2C and 2D, respectively). Thus, LuxT does not repress *luxO* transcription. By contrast, and consistent with the results in Fig 2B, WT *V. harveyi* possessed 7-fold more *luxC* mRNA than did Δ*luxT V. harveyi* at LCD (Fig 2C) while the difference was only 2-fold at HCD (Fig 2D). Thus, LuxT activates *luxCDABE* expression, primarily at LCD. Finally, measurements of AphA and LuxR protein levels showed no significant differences between the WT and Δ*luxT* strains at either LCD or HCD (Fig 2E). Because *aphA*

and *luxR* lie downstream of LuxO in the QS circuit, changes in LuxO levels necessarily drive changes in AphA and LuxR levels, albeit in opposite directions (Fig 1 and [8, 23, 28]). We conclude that LuxT has no role in regulating *luxO* expression. Therefore, LuxT activation of light production must occur through an alternative mechanism. We return to this point below.

## LuxT represses *qrr*1, not *luxO*, transcription

As mentioned in the Introduction, at the time of the Lin *et al.* studies, the Qrr sRNAs that function between LuxO and QS target genes had not been discovered. Thus, Lin *et al.* could not have known that *qrr*1 lies immediately upstream and in the opposite orientation of *luxO* in the *V. harveyi* genome. In fact, *qrr*1 is located in closer proximity to the identified LuxT binding region than *luxO*. Specifically, if +1 designates the *qrr*1 transcriptional start site, the LuxT DNA binding region spans bases -76 to -27, suggesting that LuxT binds in the *qrr*1 promoter between the predicted LuxO-P and $\sigma^{54}$ binding sites that are essential for activation of *qrr*1 transcription (Figs 3A and S2) [19, 20, 31].

To test our prediction that LuxT represses *qrr*1 transcription, not *luxO* transcription, we employed two fluorescent reporters. First, we constructed a *qrr*1 promoter fusion containing the 193 nucleotides immediately upstream of *qrr*1 fused to *mRuby3*. Thus, the promoter fragment harbored the LuxO-P, LuxT, and $\sigma^{54}$ binding sites. A consensus ribosome-binding site was included to drive *mRuby3* translation. Second, a *luxO* promoter fusion was constructed by cloning the same 193 bp DNA fragment in the opposite orientation upstream of *mRuby3*. Reporter fluorescence was measured in four *V. harveyi* strains: WT, *luxO* D61E, Δ*luxT*, and *luxO* D61E Δ*luxT*. As mentioned, *V. harveyi luxO* D61E encodes a LuxO-P mimetic. LuxO D61E constitutively activates *qrr*1-5, causing strains harboring this mutant allele to display a "LCD-locked" phenotype irrespective of the actual culture cell density [18]. The *V. harveyi luxO* D61E strain is a crucial tool for our studies. It enables investigation of the consequences of maximal *qrr* transcription when the culture cell density is high enough to allow accurate measurements of QS-controlled gene expression using reporter assays or qRT-PCR [19, 20]. The output of the P$_{qrr1}$-*mRuby3* reporter was low in the WT, *luxO* D61E, and Δ*luxT V. harveyi* strains (Fig 3B). This result was expected because *qrr*1 exhibits only low-level expression in *V. harveyi*, even at LCD [19]. Eight-fold higher expression of P$_{qrr1}$-*mRuby3* occurred in the *luxO* D61E Δ*luxT V. harveyi* strain (Fig 3B). Regarding the P$_{luxO}$-*mRuby3* reporter, compared to the WT, the output was lower in the *V. harveyi* strains harboring *luxO* D61E (Fig 3C). This result was also expected because a negative feedback loop exists between LuxO-P and *luxO* [28]. What is crucial is that elimination of *luxT* caused no change in P$_{luxO}$-*mRuby3* reporter expression compared to WT and caused no further change in the *luxO* D61E mutant (Fig 3C). Together, the *qrr*1 and *luxO* reporters show that LuxT does not regulate *luxO*. Rather, LuxT represses *qrr*1 transcription.

The distinct level of *in vivo* expression displayed by each *qrr* gene in *V. harveyi* has been interpreted to suggest that, beyond being controlled by LuxO-P, each *qrr* gene is controlled independently by other regulators [19]. Fig 3B shows that LuxT is one such regulator of *qrr*1. To investigate whether LuxT also regulates *qrr*2-5, levels of all five Qrr sRNAs were measured using qRT-PCR in WT, Δ*luxT*, *luxO* D61E, and *luxO* D61E Δ*luxT V. harveyi* strains. Confirming the reporter assay results, Qrr1 levels were ~4 fold higher in the *luxO* D61E Δ*luxT* strain than in the other three strains (Fig 3D). While increased levels of Qrr2-5 were detected in the *luxO* D61E strain compared to WT, deletion of *luxT* did not cause any additional changes (Fig 3D). Verification of the qRT-PCR results comes from analyses of *mRuby3* transcriptional reporters to *qrr*2-5. All four reporters displayed higher activity in the *luxO* D61E *V. harveyi* strain than in WT, and deletion of *luxT* had no effect (S3 Fig). Therefore, among the *qrr* genes, LuxT exclusively represses *qrr*1.

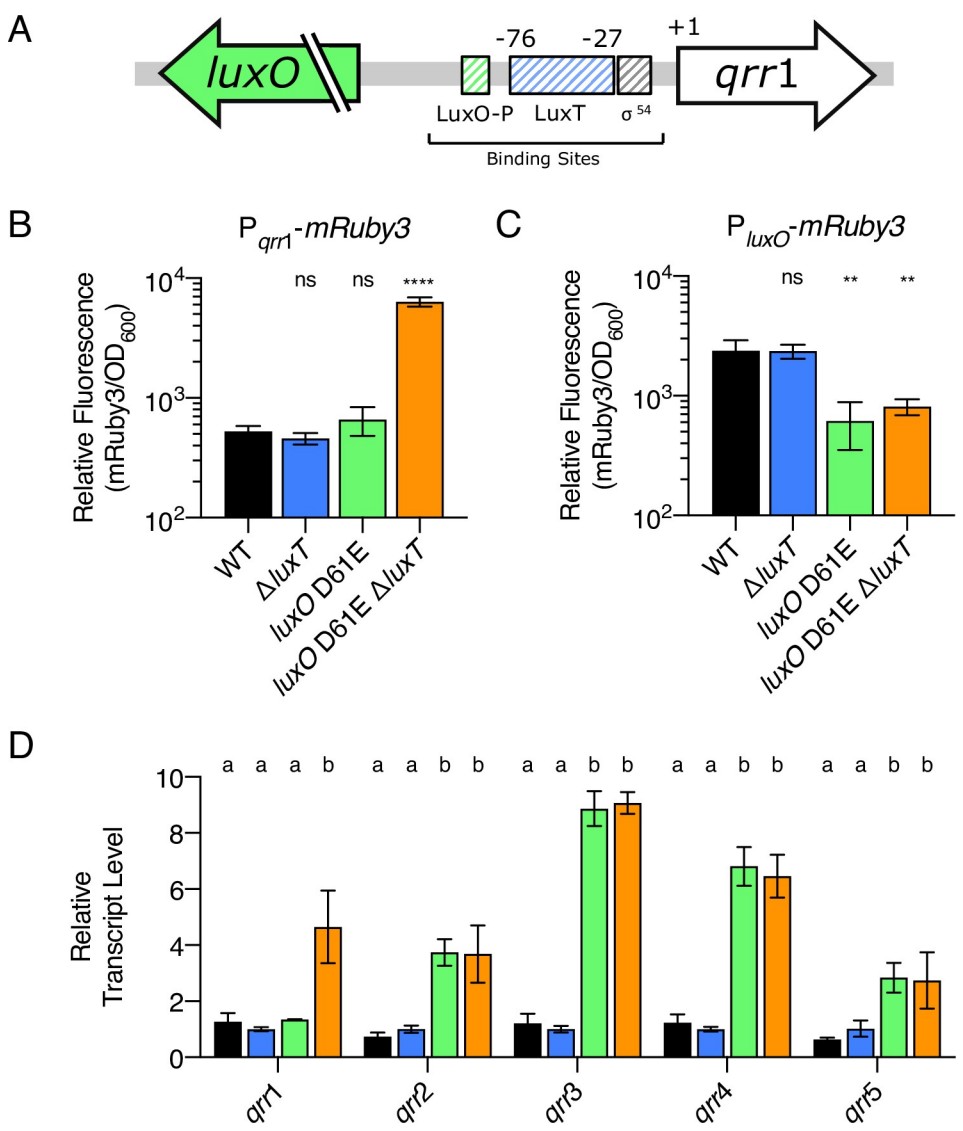

**Fig 3. LuxT represses *qrr*1 transcription. (A)** Diagram of the *luxO-qrr*1 locus. *qrr*1 resides 151 bp upstream of *luxO* and is transcribed in the opposite direction. The striped green and gray boxes depict the putative LuxO-P and $\sigma^{54}$ binding sites, respectively. The striped blue box designates the previously identified LuxT binding region, which spans from -76 to -27 relative to the *qrr*1 +1 transcriptional start site. **(B)** Relative fluorescence values (mRuby3/OD$_{600}$) of the indicated *V. harveyi* strains carrying a P$_{qrr1}$-*mRuby3* transcriptional reporter on a plasmid. Values represent relative fluorescence at OD$_{600}$ = 0.6. **(C)** As in B for strains harboring a P$_{luxO}$-*mRuby3* reporter. For B and C, unpaired two-tailed *t* tests with Welch's correction were performed comparing mutants to WT. *p* values: ns $\geq$ 0.05, ** < 0.01, **** < 0.0001. **(D)** qRT-PCR measuring the indicated Qrr sRNAs at OD$_{600}$ = 1. Transcripts were measured in WT (black), Δ*luxT* (blue), *luxO* D61E (green), and *luxO* D61E Δ*luxT* (orange) *V. harveyi*. Different letters indicate significant differences between strains, *p* < 0.05 (two-way analysis of variation (ANOVA) followed by Tukey's multiple comparisons test). For B, C, and D, error bars represent standard deviations of the means of *n* = 3 biological replicates.

## LuxT activates *luxCDABE* via a mechanism that is independent of Qrr1

Our next goal was to investigate how LuxT activates expression of *luxCDABE*, given that the mechanism is not via repression of *luxO*. The Qrr sRNAs repress *luxR* translation, and therefore they indirectly repress *luxCDABE* (Fig 1). Thus, an obvious possibility is that LuxT repression of *qrr*1 activates luciferase. However, *luxR* is downstream of Qrr1 (Fig 1) and Fig 2E

shows that deletion of *luxT* does not significantly alter LuxR levels at LCD, suggesting that LuxT does not control luciferase via a LuxR-dependent mechanism. To validate this finding, we tested whether Qrr1 is required for LuxT to activate light production. To do this, we measured bioluminescence from a *V. harveyi* Δ*luxT* mutant and compared it to that made by a Δ*luxT* Δ*qrr*1 double mutant. Both strains exhibited the identical phenotype: ~10-fold reduced light production relative to WT *V. harveyi* and the Δ*qrr*1 mutant (S4 Fig). Thus, LuxT activation of luciferase occurs by a mechanism that is independent of Qrr1.

We next tested the possibility that LuxT directly activates *luxCDABE* transcription. The *luxCDABE* promoter and regulatory region extend approximately 350 bp upstream of the *luxC* start codon [34–36]. To determine if LuxT binds within this region, we amplified six overlapping DNA fragments from -405 to +81 relative to the *luxC* start codon (S5A Fig). Compared to the avid binding of LuxT to the *qrr*1 promoter (Fig 2A), LuxT bound the *luxC* promoter only very weakly. Specifically, binding to all the *luxC* promoter-containing DNA fragments was comparable to the binding of LuxT to control (*E. coli lacZ)* DNA (Figs 2A and S5B–S5D) with modestly stronger binding to Probe 3 (S5C Fig). As another test for direct LuxT activation of luciferase, we introduced plasmid-borne arabinose-inducible *luxT* and a plasmid with IPTG-inducible *luxR* into recombinant *E. coli* carrying *luxCDABE*. LuxR is a direct activator of *lux-CDABE* [21, 36, 37]. As expected, induction of *luxR* drove increased light production compared to the empty vector control (S6A Fig). By contrast, induction of *luxT* did not increase light production in the presence or absence of *luxR* (S6A Fig). We confirmed that *luxT* was expressed from the plasmid using qRT-PCR (S6B Fig). We note that induction of *luxT* expression in *E. coli* caused a modest growth defect (S6C Fig). In conclusion, we find no evidence that LuxT directly activates *luxCDABE*.

To further investigate the mechanism underlying LuxT activation of luciferase, we probed whether LuxT functions via other known QS components. To do this, we compared the bioluminescence profiles of the *V. harveyi* Δ*qrr*1-5, Δ*luxO*, and *luxO* D61E strains to the identical strains lacking *luxT* (S7A–S7C Fig). We also included a test of the *VIBHAR_RS03920* gene (S7D Fig), a homolog of *Vibrio parahaemolyticus swrZ*. In *V. parahaemolyticus*, SwrT, the LuxT equivalent, represses *swrZ* encoding a GntR family transcription factor, which in turn, represses lateral flagellar (*laf*) genes [38]. We considered that in *V. harveyi*, LuxT could repress *VIBHAR_RS03920*, which could repress *luxCDABE*. In all four cases, introduction of the *luxT* deletion reduced light output (S7A–S7D Fig). Thus, LuxT activates *luxCDABE* by a mechanism that does not require *qrr*1-5, *luxO*, or *VIBHAR_RS03920*. We could not perform a similar experiment to assess whether LuxT regulation of *luxCDABE* is LuxR-dependent because the Δ*luxR* mutant makes no light. However, as mentioned above, LuxR protein levels are similar in WT and Δ*luxT V. harveyi* (Fig 2E), and moreover, there are no significant differences in *luxR* or *aphA* transcript levels between WT and Δ*luxT V. harveyi* at LCD (S8 Fig). Thus, LuxT affecting *luxCDABE* expression via regulation of *luxR* does not seem a reasonable possibility. To conclude, unfortunately, we did not discover the mechanism by which LuxT activates luciferase. We do know that the mechanism is likely indirect and that the component that connects LuxT to *luxCDABE* is not any of the regulators in the *V. harveyi* QS pathway. From here forward, we focus on the consequences of LuxT regulation of *qrr*1. In future studies, we hope to define the mechanism by which LuxT activates light production.

## LuxT controls target genes via repression of *qrr*1

Only low-level expression of *qrr*1 occurs in WT *V. harveyi*, including at LCD, and that feature has made it difficult to detect Qrr1-mediated regulatory effects *in vivo*. Based on our discovery of LuxT repression of *qrr*1, we hypothesize that LuxT activity could mask Qrr1 function *in*

*vivo*. If so, LuxT would indirectly activate the known Qrr1-repressed mRNA targets. To test this possibility, we used qRT-PCR to compare the levels of Qrr1 mRNA targets in *V. harveyi luxO* D61E to that in *V. harveyi luxO* D61E Δ*luxT*. We assayed the 14 Qrr1 target genes that lie outside the QS pathway [30] as well as *luxR* and *luxMN*, Qrr1 targets that function inside the QS system [19, 26, 29]. Deletion of *luxT* caused a significant decrease in the mRNA levels of 9 of the 16 tested genes (S9A Fig). Thus, we suspected that LuxT activated expression of the 9 genes via repression of *qrr1*. To test this prediction, we compared transcript levels of the 9 genes in *V. harveyi luxO* D61E, *V. harveyi luxO* D61E Δ*luxT*, *V. harveyi luxO* D61E Δ*qrr1*, and *V. harveyi luxO* D61E Δ*qrr1* Δ*luxT*. To our surprise, in all cases, the two strains lacking *luxT* possessed lower levels of the transcripts than did the two strains possessing *luxT* (S9B Fig). These data show that these target genes are controlled by LuxT in a Qrr1-independent manner.

The data in S9B Fig inspired us to expand our LuxT/Qrr1 regulatory model to include two key findings: (1) LuxT represses *qrr1*, encoding a sRNA that post-transcriptionally regulates target genes (Fig 3 and [30]), and (2) LuxT also activates expression of the same target genes, independently of Qrr1. Thus, we propose that LuxT functions by two mechanisms to activate expression of the 9 target genes, one transcriptionally and one post-transcriptionally: LuxT is a transcriptional activator of the target genes and LuxT additionally activates the target genes by repressing their repressor, Qrr1.

To test the above model, we focused on the four most highly LuxT-regulated target genes: *VIBHAR_RS11785*, *VIBHAR_RS11620*, *VIBHAR_RS16980*, and *VIBHAR_RS25670*. First, to examine whether LuxT indeed activates their transcription, we eliminated Qrr-dependent regulation using a *V. harveyi* Δ*qrr1-5* strain. In all four cases, transcript levels were lower in the Δ*qrr1-5* Δ*luxT* strain than in the Δ*qrr1-5* strain. Complementation with *luxT* expressed from a plasmid restored the transcript levels, confirming that LuxT activates the expression of these genes via a Qrr-independent mechanism (Fig 4A). To demonstrate that LuxT control of these genes is exerted at the level of transcription, we made *lux* transcriptional reporters and measured their outputs in *luxA*::Tn5 and *luxA*::Tn5 Δ*luxT V. harveyi* strains. Using a *luxA* null mutant for this analysis ensured that all light production came from the transcriptional fusions. All four reporters exhibited lower activity in the *luxA*::Tn5 Δ*luxT* strain than in the *luxA*::Tn5 strain (~400, 4, 48, and 7-fold lower activity for, respectively, *VIBHAR_RS11785*, *VIBHAR_RS11620*, *VIBHAR_RS16980*, and *VIBHAR_RS25670*, S10 Fig). These data confirm an aspect of our model: LuxT activates transcription of these target genes.

The other tenet of our model, that LuxT activates expression of the target genes via repression of *qrr1* cannot be detected by the above qRT-PCR assay (S9B Fig). Fig 4B and 4C depicts the issue. In WT *V. harveyi*, transcription of *qrr1* is repressed by LuxT. Therefore, deletion of *qrr1* has no effect on target gene regulation (Figs 4B and S9B). In Δ*luxT V. harveyi*, *qrr1* expression is de-repressed. However, in the absence of the LuxT activator, transcription of the target genes does not occur. Thus, although Qrr1 is present, its mRNA targets are absent, so again regulation via Qrr1 does not occur (Fig 4C).

To circumvent these issues and probe the connection of LuxT to Qrr1 in post-transcriptional regulation of the four target genes, we used a strategy in which we eliminated LuxT transcriptional control of the target genes to unmask post-transcriptional effects. To accomplish this, we constructed translational fusions to the fluorescent protein mVenus. DNA upstream of each target gene containing the site that base pairs with Qrr1 and the ribosome-binding site was cloned in frame with *mVenus* downstream of the tetracycline-inducible *tetA* promoter. Therefore, the fusions were constitutively transcribed following addition of aTc, irrespective of the presence or absence of *luxT*. Analogously designed translational reporters were previously shown to be repressed in *E. coli* following *qrr1* overexpression [30]. We confirmed that the

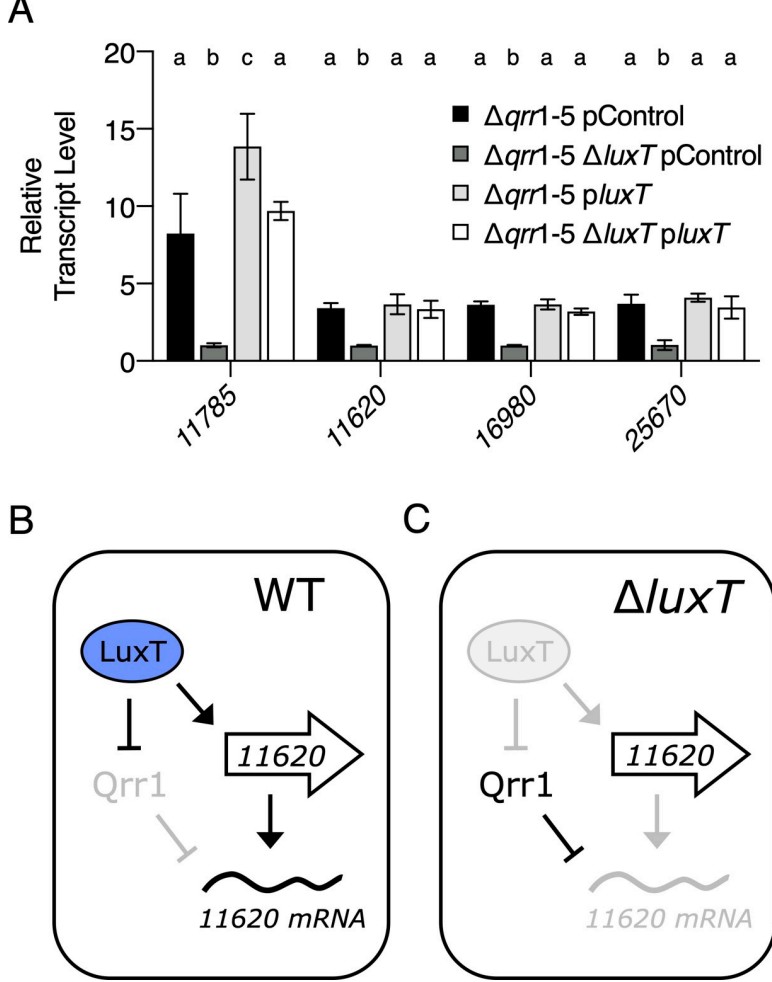

**Fig 4. LuxT activates target genes by two regulatory mechanisms. (A)** qRT-PCR of the indicated *VIBHAR_RS* genes in the designated *V. harveyi* strains. The pControl plasmid is the empty parent vector and the plasmid designated p*luxT* carries *luxT* under the IPTG-inducible *tac* promoter. In all cases, 0.5 mM IPTG was added and samples were collected at $OD_{600} = 1$. Error bars represent standard deviations of the means of $n = 3$ biological replicates. Different letters indicate significant differences between strains, $p < 0.05$ (two-way analysis of variation (ANOVA) followed by Tukey's multiple comparisons test). **(B)** Working model for how LuxT activates a target gene, with *VIBHAR_RS11620* as the example in WT *V. harveyi*. **(C)** As in B for Δ*luxT V. harveyi*.

reporters are all activated by aTc and repressed following overexpression of *qrr*1 in *V. harveyi* (S11 Fig).

The translational mVenus reporter fusions were used to test the aspect of our model in which we predict that LuxT activates target genes post-transcriptionally via *qrr*1 repression. Reporter activities from the four target gene constructs were measured in the following LCD-locked *V. harveyi* strains: *luxO* D61E, *luxO* D61E Δ*luxT*, *luxO* D61E Δ*qrr*1, and *luxO* D61E Δ*qrr*1 Δ*luxT*. The results for all four reporters were similar (Fig 5A–5D). The *luxO* D61E strain exhibited higher reporter activity than the *luxO* D61E Δ*luxT* strain, presumably due to the de-repression of *qrr*1 that occurs in the absence of LuxT. Importantly, deletion of *luxT* in the *luxO* D61E Δ*qrr*1 strain had no effect on reporter translation (Fig 5A–5D, compare *luxO* D61E Δ*qrr*1 and *luxO* D61E Δ*qrr*1 Δ*luxT* bars). We conclude that LuxT post-transcriptionally regulates the four tested genes in a Qrr1-dependent manner. We note that higher translation

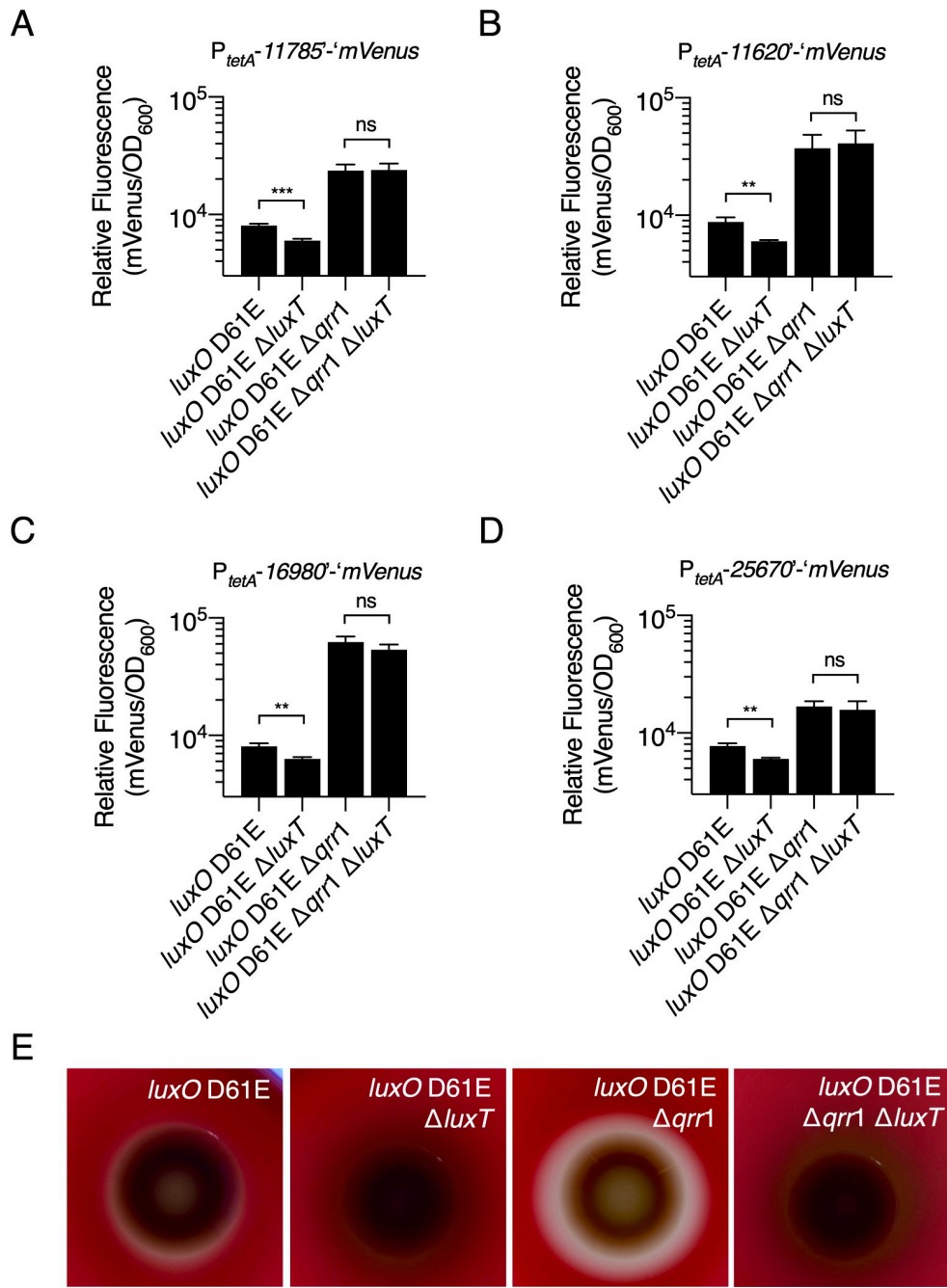

**Fig 5. LuxT post-transcriptionally activates target genes via repression of *qrr*1. (A-D)** Relative fluorescence values (mVenus/OD$_{600}$) of the indicated *V. harveyi* strains harboring plasmids carrying translational mVenus reporters to the indicated genes. In all cases, 100 ng mL$^{-1}$ aTc was added to induce constitutive transcription of the reporters from the *tetA* promoter. Values represent relative fluorescence at OD$_{600}$ = 0.3 for each sample. Error bars represent standard deviations of the means of *n* = 3 biological replicates. Unpaired two-tailed *t* tests with Welch's correction were performed comparing two samples, as indicated. *p*-values: ns $\geq$ 0.05, ** $<$ 0.01, *** $<$ 0.001. **(E)** Halo formation by the indicated *V. harveyi* strains on TSA plates containing 5% sheep's blood. Plates were incubated at 30°C for 72 h. A zone of clearing surrounding the colony indicates aerolysin-driven hemolysis.

of the reporters occurred in the *luxO* D61E Δ*qrr*1 strains than the *luxO* D61E strains containing *qrr*1 (Fig 5A–5D). This pattern is consistent with Qrr1 functioning as a repressor, and we interpret the result to mean that when the *qrr*1 gene is present, residual Qrr1 production occurs, including in the presence of LuxT. We presume that this pattern cannot be observed in the qRT-PCR analyses (S9B Fig) because Qrr1 represses translation of target genes by a sequestration mechanism that does not significantly alter mRNA levels [25, 30].

The four genes that are regulated transcriptionally by LuxT and post-transcriptionally by LuxT via Qrr1 encode a peptidase (*VIBHAR_RS11785*), an aerolysin toxin (*VIBHAR_RS11620*), a chitin disaccharide deacetylase (*VIBHAR_RS16980*), and a protein involved in export of capsular polysaccharide (*VIBHAR_RS25670*). Interestingly, all four genes encode secreted public goods or products involved in secretion of public goods (i.e., *VIBHAR_RS25670*), a class of components that are commonly controlled by QS. We focus on the aerolysin toxin (*VIBHAR_RS11620*) here to probe *in vivo* LuxT and Qrr1 regulation. Secreted aerolysin-like toxins form pores in eukaryotic cells, and in the case of red blood cells, cause lysis [39]. Thus, aerolysin hemolytic activity can be assessed by growing bacteria on blood agar plates and monitoring them for zones of clearance. We used this assay to test if LuxT and Qrr1 influence aerolysin secretion according to our dual-mechanism model (Fig 4B). First, the *V. harveyi luxO* D61E strain exhibited modest clearing, whereas no clearing occurred around the *luxO* D61E Δ*luxT* strain (Fig 5E). This result is consistent with LuxT functioning as an activator of aerolysin production. Second, compared to the *luxO* D61E strain, *luxO* D61E Δ*qrr*1 showed increased hemolytic activity (Fig 5E). This result can be explained by Qrr1-mediated post-transcriptional repression of *VIBHAR_RS11620* (Fig 5B). Finally, the *luxO* D61E Δ*qrr*1 Δ*luxT* strain did not display hemolytic activity (Fig 5E). In agreement with our model (Fig 4B and 4C), the transcriptional effect of LuxT overrides the post-transcriptional effect of Qrr1. The hemolysis activities of the identical strains were also quantified using a liquid assay (S12 Fig). Analogous results were obtained for the four strains except that the *luxO* D61E strain exhibited a level of hemolytic activity similar to that of the *luxO* D61E Δ*qrr*1 strain. Possibly, this discrepancy is due to the different growth conditions used for the plate and liquid hemolysis assays.

## LuxT represses *qrr*1 in *A. fischeri*

In members of the *Vibrionaceae* family, AI structures and the types of proteins employed as receptors vary between species. However, LuxO is conserved in all sequenced vibrio species [40] and LuxT is also often present [38, 41–43] and we address this further in the next section. We wondered whether LuxT-mediated repression of *qrr*1 is *V. harveyi* specific or whether LuxT has this function in other *Vibrionaceae* species. To explore this question, we tested three species, *Vibrio cholerae*, *V. parahaemolyticus*, and *A. fischeri* in experiments analogous to those in Fig 3B. Plasmids harboring transcriptional reporter fusions to *qrr*1 from each representative species were introduced into WT, Δ*luxT*, *luxO* D61E, and *luxO* D61E Δ*luxT* strains of those species. As mentioned, *luxT* is called *swrT* in *V. parahaemolyticus*, and the LCD-locked LuxO-P mimetic in *A. fischeri* is *luxO* D55E. In *V. cholerae*, LuxO D61E activated the P$_{qrr1}$-*luxCDABE* reporter relative to WT, however elimination of *luxT* did not affect reporter activity in either strain (S13A Fig). Activity from the *V. parahaemolyticus* P$_{qrr1}$-*mRuby3* reporter remained low in all four strains (S13B Fig). Thus, we do not find evidence for *qrr*1 repression by LuxT in *V. cholerae* or by SwrT in *V. parahaemolyticus*. We note, however, that regarding *V. parahaemolyticus*, we cannot rule out the presence of an additional *qrr*1 repressor that masks LuxT function and maintains *qrr*1 transcription at an especially low level.

*A. fischeri* is distantly related to *V. harveyi* and, curiously, *A. fischeri* only encodes a single *qrr* gene, *qrr*1, and Qrr1 post-transcriptionally represses LitR, the LuxR homolog (Fig 6A)

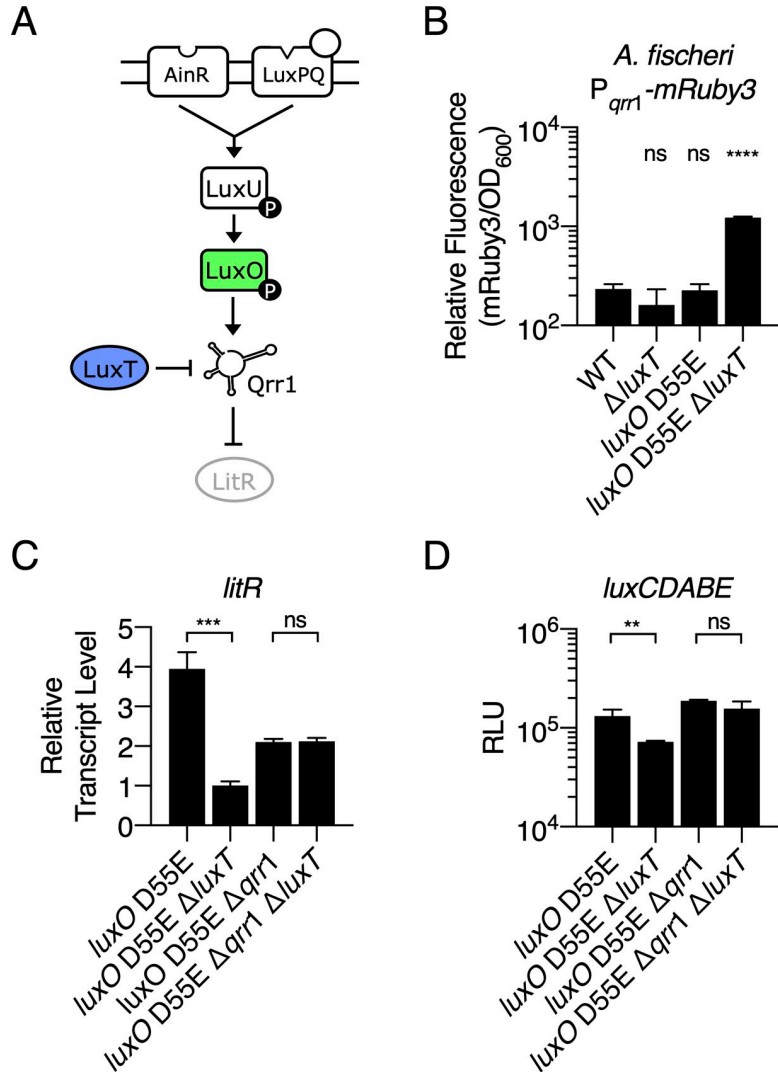

**Fig 6. LuxT represses *qrr*1 in *A. fischeri*.** **(A)** Simplified *A. fischeri* QS pathway at LCD. See text for details. **(B)** Relative fluorescence values (mRuby3/OD$_{600}$) of the indicated *A. fischeri* strains carrying a P$_{qrr1}$-*mRuby3* transcriptional reporter on a plasmid. Values represent relative fluorescence at OD$_{600}$ = 0.6 for each sample. **(C)** *litR* mRNA levels in the designated *A. fischeri* strains at OD$_{600}$ = 1 obtained by qRT-PCR. **(D)** Bioluminescence production of the indicated *A. fischeri* strains at OD$_{600}$ = 1. Relative light units (RLU) are counts/min mL$^{-1}$ per OD$_{600}$. For B, C, and D, error bars represent standard deviations of the means of $n$ = 3 biological replicates, and unpaired two-tailed $t$ tests with Welch's correction were conducted comparing the WT to the mutants (B) or the two indicated samples (C and D). $p$-values: ns $\geq$ 0.05, ** < 0.01, *** < 0.001, **** < 0.0001.

[44]. Through additional regulatory steps, activation of LitR drives the downstream activation of *luxCDABE* [45]. The *A. fischeri* P$_{qrr1}$-*mRuby3* reporter exhibited low-level expression in the WT, Δ*luxT*, and *luxO* D55E strains (Fig 6B). However higher fluorescence was emitted in the *A. fischeri luxO* D55E Δ*luxT* strain (Fig 6B). Thus, as in *V. harveyi*, LuxT is a repressor of *qrr*1 in *A. fischeri*.

The redundancy among the five Qrr sRNAs in *V. harveyi* prevents the elimination of *qrr*1 from driving large effects on LuxR levels (Fig 2E and [19]), and in the context of the present work, masks the consequences of deletion of *luxT*. Because no Qrr redundancy exists in *A. fischeri*, we predicted that LuxT repression of *qrr*1 would affect LitR levels. Indeed, compared

to the *A. fischeri luxO* D55E strain, the *luxO* D55E Δ*luxT* strain showed a 4-fold reduction in *litR* transcript levels (Fig 6C). To test if this manifestation of LuxT occurs via repression of *qrr*1, we measured *litR* transcription in *A. fischeri luxO* D55E Δ*qrr*1 and *A. fischeri luxO* D55E Δ*qrr*1 Δ*luxT*. There was no significant difference in *litR* transcript levels showing that LuxT activates *litR* expression in a Qrr1-dependent manner (Fig 6C). The differences in *litR* transcript levels observed between the *luxO* D55E and *luxO* D55E Δ*qrr*1 strains are likely a result of Qrr1 feedback control of *luxO* [28]. To determine if the observed LuxT-dependent effects on LitR likewise affect downstream expression of luciferase, we measured bioluminescence in the four *A. fischeri* strains. Indeed, the *luxO* D55E Δ*luxT* strain made less light than the *luxO* D55E strain (Fig 6D). The *luxO* D55E Δ*qrr*1 and *luxO* D55E Δ*qrr*1 Δ*luxT* strains emitted similar levels of light showing that LuxT controls light production in *A. fischeri* via regulation of *qrr*1 (Fig 6D). We conclude that LuxT is a repressor of *qrr*1 in *A. fischeri*, and because Qrr1 is the sole Qrr, LuxT has a more major role in controlling the overall QS state in *A. fischeri* than in *V. harveyi*. We discuss possible advantages of the different regulatory arrangements below.

## Putative LuxT regulation of *qrr*1 is diversified in the *Vibrionaceae* family

Members of the *Vibrionaceae* family can be divided into two classes, those encoding a single *qrr* upstream of *luxO*, and those encoding multiple *qrr* loci [20, 44]. Species with multiple *qrr* genes always encode *qrr*1 upstream of *luxO*, suggesting that *qrr*1 is the ancestral gene. Our finding of LuxT repression of *qrr*1 in both *V. harveyi* and *A. fischeri* inspired us to investigate whether LuxT is conserved among all *Vibrionaceae* family members, and if so, whether LuxT possesses an evolutionary pattern that corresponds to that of the Qrr sRNAs. To compare *luxT* and *qrr* phylogenies, we scanned all *Vibrionaceae* sequenced genomes to identify *qrr* genes, expanding on previous analyses [19, 20, 44]. The majority of species within the *Vibrio* genus encoded multiple *qrr* loci, most often 4 or 5 *qrr* genes, like *V. cholerae* and *V. harveyi*, respectively (Fig 7A). All members of non-*Vibrio* genera encoded only a single *qrr* gene, like *A. fischeri*, except for *Photobacterium galatheae*, which had no putative *qrr* gene (Fig 7A). Analogous examination of the genomes for *luxT* homologs showed that *luxT* genes exist in most *Vibrionaceae* species possessing one and multiple *qrr* genes (Fig 7B). Within the *Vibrio* genus, species lacking apparent *qrr* genes also lacked *luxT* homologs, and the *luxT* genes were more similar to *V. harveyi luxT* in species with multiple *qrr* genes than were the *luxT* genes in species possessing only a single *qrr* gene.

To predict whether LuxT does or does not control *qrr*1 expression in a particular species, we compared the DNA sequences upstream of *qrr*1 in the four *Vibrionaceae* species tested in our experiments. The σ54 binding sites are highly conserved among the four species (Fig 7C and [19, 20]), while the LuxT binding regions show less conservation. Thus, harboring a *luxT* homolog does not necessarily signify that it controls *qrr*1. The "GGTTAAA" upstream of *qrr*1 in the LuxT binding region was the most conserved sequence between the species. Consistent with our experimental results, the *V. cholerae* sequence in this region, i.e., "GATTTG–", is the most dissimilar from those of the other three species (Fig 7C). This sequence divergence may underlie our finding that LuxT does not regulate *qrr*1 in *V. cholerae* (S13 Fig). In *V. parahaemolyticus*, this region is identical to that in *V. harveyi*. However, we do not observe LuxT regulation of *qrr*1 in *V. parahaemolyticus* (S13 Fig). As mentioned above, *qrr*1 expression in *V. parahaemolyticus* may be too low to detect repression by LuxT, possibly due to additional repression by another factor. To more broadly examine the conservation of LuxT binding regions, we also performed phylogenetic analysis comparing the putative LuxT binding regions in the *qrr*1 promoters of all *Vibrionaceae* family members possessing both *qrr*1 and *luxT* genes. A variety of sequences exist (Fig 7D), and we find no evidence for a correlation

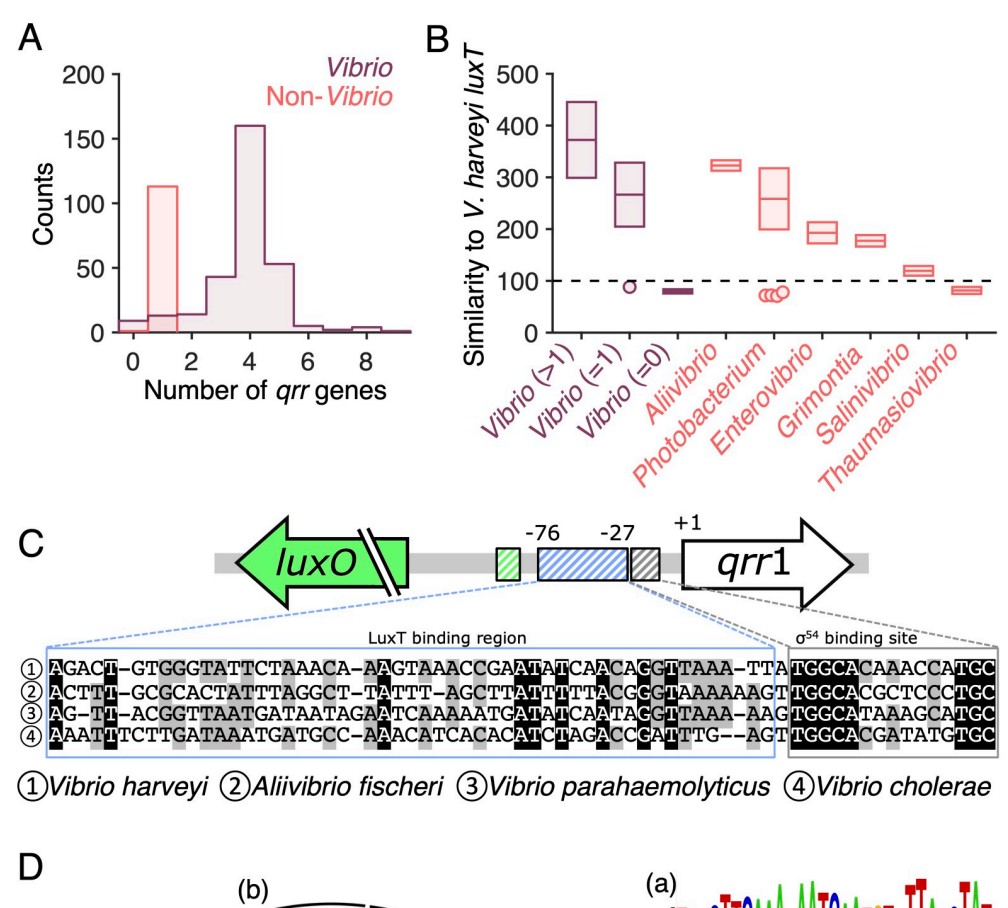

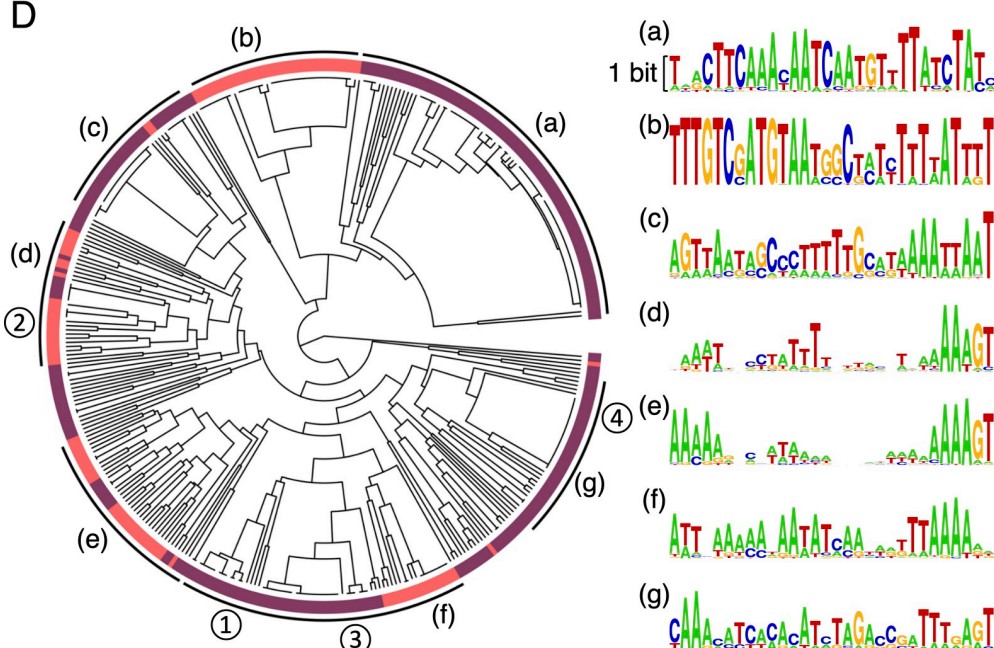

**Fig 7. Co-occurrence of *luxT* and *qrr* genes and possible LuxT regulation of *qrr*1 across the *Vibrionaceae*. (A)**
Histogram of the number of *qrr* genes in *Vibrio* (purple) and non-*Vibrio* (red) members of the *Vibrionaceae* family.
**(B)** Highest similarity score to *V. harveyi luxT* for genes in the indicated genera. The vibrios are divided into three
groups based on the number of *qrr* genes in their genomes (indicated by the numbers in the parentheses). The
similarity scores, which quantify the weighted DNA sequence similarities based on the standard scoring matrix
NUC44, were obtained from alignments of genome sequences to the query probe using the Smith-Waterman
algorithm (see Methods). The black dashed line indicates the cutoff used for the similarity score. Boxes show the

means ± SD. Circles represent outlier species whose highest similarity scores to *V. harveyi luxT* fell below the cutoff. (**C**) Alignment of *qrr*1 upstream DNA sequences for the indicated species. Gray and black denote 75% and 100% consensus, respectively. The σ[54] binding site and the LuxT binding region are indicated. Colors as in Fig 3A. (**D**) Phylogenetic tree of *Vibrionaceae* family members based on the 30 nucleotides upstream of the σ[54] binding sites in the *qrr*1 promoters. Colors as in panels A and B. Branches corresponding to species shown in panel C are indicated by the circled numbers. Groups of species with highly similar upstream sequences (sequence logos shown on the right) are indicated by letters in parentheses. Regarding the sequence logos, the heights of the different nucleotides are scaled according to their frequencies at each position, and the height of each nucleotide stack is proportional to the information content (measured in bits) of the corresponding position. Scale bar, 1 bit.

between the number of *qrr* genes and similarity in the upstream LuxT binding regions. It remains possible that the DNA binding domains of LuxT coevolve with the DNA sequences in the LuxT binding regions. Together, our results indicate that while *qrr*1 and *luxT* are broadly conserved in *Vibrionaceae* species, LuxT regulation of *qrr*1 has diversified. Going forward, we will combine experimental and bioinformatic approaches to pinpoint the precise LuxT binding site, determine its conservation between species, and define the ramifications of particular DNA sequence changes.

## Discussion

To survive, bacteria must appropriately respond to fluctuating environments. For marine bacteria such as *V. harveyi*, successfully competing against a diversity of other microbes and adapting to dynamic microscale nutrient gradients are key [46, 47]. Sensory relays that tune gene expression via transcriptional and post-transcriptional mechanisms enable bacteria to overcome varying environmental challenges [48]. In the context of the present work, QS signal transduction allows bacteria to monitor their changing cell numbers and transition between executing individual and collective activities [49].

In vibrios, one or more Qrr sRNAs function at the core of QS signaling pathways, and thus the concentration of Qrr sRNAs present at any time dictates the QS output response in which hundreds of genes are either activated or repressed. The Qrr sRNAs, and other bacterial sRNAs, are post-transcriptional regulators. Bacterial sRNAs are thought to be especially beneficial regulators due to the low metabolic cost of their production coupled with their fast synthesis and turnover rates, the latter of which can drive rapid changes in target mRNA levels [50, 51]. Moreover, because the QS Qrr sRNAs function by multiple mechanisms (sequestration, catalytic mRNA degradation, coupled mRNA-sRNA degradation, and mRNA translational activation), they can confer distinct timing and expression levels to particular target genes providing "bespoke" QS output responses [25]. These features of sRNAs are presumed to drive dynamic patterns of gene expression that might not be achievable through the use of canonical transcription factors.

Gene duplication has led to the *V. harveyi* QS circuit harboring five similar Qrr sRNAs [19]. Beyond QS, in bacteria it is common for multiple sRNAs to function redundantly in a single pathway. Presumably, possessing more than one copy of a sRNA gene can increase the available sRNA pool, and in turn, confer increased control over target gene expression. In addition or alternatively, duplication may allow individual sRNA genes to diversify, in sequence and/or in expression pattern, either or both of which can enable differential regulatory effects [52]. Indeed, regarding the *V. harveyi* Qrr sRNAs, deletion analyses and Qrr quantitation studies have demonstrated that the pool of Qrr sRNAs available to regulate downstream target gene expression increases with increasing numbers of *qrr* genes. Curiously, however, at least in the laboratory and with *luxR* as the measured target gene, only four of the five Qrr sRNA genes are required to achieve this effect. Thus, the final *qrr* duplication event

does not appear to enhance regulatory control [19]. Moreover, only low-level production of Qrr1 and Qrr5 have been documented, suggesting that those two sRNAs do not contribute dramatically to changes in the levels of the sRNA pool [19]. These findings, together with the knowledge that the *qrr* promoter regions vary, has led us to hypothesize that some or all of the *qrr* genes may be subject to additional control by as yet undefined regulatory components.

Here, our discovery of *V. harveyi* LuxT as a repressor of *qrr*1 provides evidence for a QS model in which individual *qrr* genes are uniquely regulated. While LuxT repression of *qrr*1 does not affect expression of the genes encoding the master QS regulators LuxR and AphA, it does alter expression of a subset of Qrr1 target genes. Separate from its role as a *qrr*1 repressor, we also found that LuxT controls the same set of Qrr1 target genes at the transcriptional level. A regulatory strategy in which control is exerted at two levels, via a transcriptional regulator and a post-transcriptional sRNA, occurs in other systems and is proposed to prevent leaky target gene expression and to alter target gene expression dynamics [53, 54]. In the case of LuxT, at least four genes are subject to such control, and they encode a protease, an aerolysin toxin, a chitin deacetylase, and a gene involved in capsular polysaccharide secretion. Notably, all four gene products are secreted, perhaps emphasizing the need for especially tight control of public goods production. We imagine that LuxT initially evolved to transcriptionally activate this set of target genes and later incorporated repression of *V. harveyi qrr*1 to reinforce activation at the post-transcriptional level. Thus, the gene duplication events that generated *qrr* redundancy in *V. harveyi* also provided the required substrate for regulation by LuxT, ultimately enabling finely tuned expression of select members of the QS regulon that rely on Qrr1, while avoiding blanket alteration of the QS response. Our discovery of LuxT repression of *V. harveyi qrr*1 hints that analogous regulators may exist that uniquely control *qrr*2-5.

The *luxT* gene is conserved among *Vibrionaceae* bacteria, but we only observe LuxT repression of *qrr*1 in two of four tested species, *V. harveyi* and *A. fischeri*. These two species harbor five and one *qrr* genes, respectively. More broadly, our phylogenetic analyses of the LuxT binding regions upstream of *Vibrionaceae qrr*1 genes show that this DNA sequence has diversified, and consistent with our results, may signify that LuxT represses only a subset of *qrr*1 genes. Further investigation is necessary to understand the regulatory logic underlying LuxT repression of *qrr*1 in some species but not in others. We can speculate on these different circuit arrangements. To do so, we consider the diversity of QS system components and regulatory architectures present in *Vibrionaceae* species. We know from our and previous phylogenetic analyses that *luxO* is highly conserved in vibrios, and species commonly possess from one to five *qrr* genes [20, 40, 44, 55]. Beyond these two core components, *Vibrionaceae* QS systems vary with respect to the number and structures of QS AIs, the number, subcellular locations, and signal relay mechanisms of the QS receptors, and the number and identities of the downstream target genes [1, 56, 57]. Presumably, the differences in QS system architectures represent the outcomes of distinct selective pressures experienced by particular species over evolutionary time. As species diverged, a common set of parts were mixed and matched, duplicated, and their placements in the regulatory hierarchies altered with LuxO and the Qrr sRNAs remaining as the core of the QS networks. Similar, but not identical QS systems emerged, each presumably capable of promoting ideal biology for a given species. With regard to the present work, LuxT represents one more component that evolution can insert into *Vibrionaceae* QS systems in different places in the various hierarchies to enable it to specialize for each species.

Lastly, LuxT is a member of the bacterial TetR family of transcriptional regulators, a widely distributed family of proteins possessing characteristic helix-turn-helix DNA-binding domains [58]. *V. harveyi* LuxR is a member of this same protein family. Prior to our discovery of *V. harveyi* LuxT as a *qrr*1 repressor, the functions of some LuxT homologs had been studied

including in *V. parahaemolyticus*, *A. fischeri*, *Vibrio vulnificus*, and *Vibrio alginolyticus*. In *V. parahaemolyticus*, the LuxT homolog, SwrT, activates genes promoting lateral-flagellar-driven swarming, enabling translocation across surfaces [38, 59–61]. LuxT is a transcriptional activator of siderophore biosynthetic genes in *A. fischeri* [43]. In *V. vulnificus* and *V. alginolyticus*, LuxT is reported to control QS via regulation of expression of the *luxR* homologs [41, 42]. Additionally, the *V. alginolyticus* Δ*luxT* mutant is defective for virulence in a zebrafish infection model [42]. Whether Qrr1 acts as a LuxT-controlled intermediary in these other vibrio pathways has not been investigated. These earlier studies, together with our findings that LuxT also controls gene expression independent of Qrr1 in *V. harveyi* hint that LuxT is a global regulator of gene expression in *Vibrionaceae*. Future transcriptomic analyses will be used to identify the set of genes comprising the *V. harveyi* LuxT regulon and to fully define which LuxT target genes are Qrr1 dependent and which are Qrr1 independent. Similar analyses in other *Vibrionaceae* species could reveal which functions of LuxT are general and which are species specific. Finally, it will be of particular interest to investigate the environmental signals that control *luxT* expression and LuxT activity. Under standard laboratory conditions, we have not observed variation in *luxT* mRNA or protein levels, however, examining its activity under conditions that more closely mimic nature may reveal how *luxT* itself is regulated.

## Materials and methods

### Bacterial strains and culture conditions

*V. harveyi* strains were derived from *V. harveyi* BB120 (BAA-1116) [62]. *A. fischeri* strains were derivatives of *A. fischeri* ES114 [63]. *V. cholerae* strains were derived from *V. cholerae* C6706str2 [64], and *V. parahaemolyticus* strains were derived from *V. parahaemolyticus* BB22OP (LM5312) [65]. *E. coli* BW25113 was used for heterologous gene expression and *E. coli* S17–1 λ*pir* was used for cloning. All strains are listed in S1 Table. *Vibrio* and *Aliivibrio* strains were grown at 30°C shaking in either Luria Marine (LM) medium or minimal Autoinducer Bioassay (AB) medium, the latter supplemented with 0.4% vitamin-free casamino acids (Difco) [4, 66]. *E. coli* strains were grown shaking at 37°C or at 30°C in LB medium. Antibiotics were added as follows (µg mL$^{-1}$): ampicillin, 100; chloramphenicol, 10; kanamycin, 100; polymyxin B, 50; and tetracycline, 10. Induction of genes on plasmids was accomplished by the addition of 0.5 mM isopropyl β-D-1-thiogalactopyranoside (IPTG) (Thermo Fisher), 0.2% arabinose (Sigma), or 100 ng mL$^{-1}$ anhydrotetracycline (aTc) (Takara), as necessary.

### DNA manipulation and strain construction

PCR reactions were carried out with either KOD Hot Start DNA Polymerase (Sigma) or iProof DNA Polymerase (Bio-Rad). Oligonucleotides were purchased at Integrated DNA Technologies (IDT) and are listed in S2 Table. A DNA fragment containing the randomized LuxT binding region was synthesized by IDT. Cloning was performed using isothermal DNA assembly with the Gibson Assembly Master Mix (New England Biolabs) [67]. All plasmids were validated by sequencing (Genewiz) and are listed in S3 Table. Plasmids that enable overexpression of genes are designated with a lowercase p (e.g. p*qrr*1). For reporter fusion constructs, a capital P designates the promoter that drives transcription (e.g. P$_{qrr1}$-*mRuby3*). Transcriptional reporters to *luxO* and to *qrr*1 included approximately 200 bp of promoter DNA upstream of *mRuby3*. Transcriptional reporters to *qrr*2-5, *VIBHAR_RS11785*, *VIBHAR_RS11620*, *VIBHAR_RS16980*, and *VIBHAR_RS25670* contained approximately 300 bp of promoter DNA. A consensus ribosome binding site was included to drive translation. The putative base pairing regions between Qrr1 and the *VIBHAR_RS11785*, *VIBHAR_RS11620*, and *VIBHAR_RS25670* mRNAs were excluded from those reporter constructs. Due to its location far upstream of the

gene, the putative Qrr1 base pairing region for *VIBHAR_RS16980* could not be excluded [30]. Translational reporters employing *mVenus* were designed using a previously described method and transcribed from the aTc inducible *tetA* promoter [30, 68]. Plasmids were introduced into *E. coli* by electroporation using a Bio-Rad Micro Pulser. Plasmids were introduced into *Vibrio* and *Aliivibrio* strains via conjugation with *E. coli* S17-1 λ*pir. V. harveyi, V. cholerae*, and *V. parahaemolyticus* exconjugants were selected on agar plates with polymyxin B. *A. fischeri* exconjugants were selected on agar plates containing ampicillin. Chromosomal alterations in *Vibrio* and *Aliivibrio* strains were generated using the pRE112 suicide vector harboring the *sacB* counter-selectable marker as previously described [34, 43, 69]. Selection for the second crossover event was performed on LM agar plates containing 15% sucrose (Sigma). Mutations were validated by PCR and/or sequencing.

## LuxT-6xHis protein production and purification

The DNA encoding LuxT-6xHis was cloned into the pET-15b vector and the protein was over-expressed in *E. coli* BL21 (DE3) using 0.4 mM IPTG at 18˚C for overnight growth. Cells were pelleted at 16,100 x *g* for 10 min and resuspended in lysis buffer (25 mM Tris-HCl pH 8, 150 mM NaCl) supplemented with 1 mM DTT, 2 mM PMSF, and 5 μM DNase I. The cells were lysed using sonication and subjected to centrifugation at 32,000 x *g* for 1 h. The LuxT-6xHis protein was purified from the clarified supernatant by Ni-NTA Superflow resin (Qiagen). Following washes with lysis buffer containing 20 mM Imidazole, the protein was eluted using lysis buffer containing 300 mM Imidazole. The collected elution fraction was loaded onto a HiTrap Q column (GE Healthcare) and further purified using a linear gradient of buffer A (25 mM Tris-HCl pH 8, 1 mM DTT) to buffer B (25 mM Tris-HCl pH 8, 1 M NaCl, 1 mM DTT). Peak fractions were pooled, concentrated, and subjected to a Superdex-200 size exclusion column (GE Healthcare) in gel filtration buffer (25 mM Tris-HCl pH 8, 100 mM NaCl, 1 mM DTT). The protein was concentrated, flash frozen, and stored at -80˚C.

## Electrophoretic mobility shift assays (EMSAs)

Oligonucleotide primers used to amplify DNA probes are listed in S2 Table. Reaction mixtures of 10 μL volume containing 20 nM dsDNA probe and 1:2 serial dilutions of LuxT-6xHis in low salt buffer (25 mM Tris pH 8, 50 mM NaCl) were incubated at room temperature for 15 min. LuxT-6xHis dimer concentrations ranged from 10 nM (0.5x) to 320 nM (16x). After incubation, 2.5 μL of 5X loading buffer (LightShift EMSA Optimization and Control Kit, Thermo) was added to the mixtures, and the samples were loaded onto a 6% Novex TBE DNA retardation gel (Thermo) at 4˚C. Gels were subjected to electrophoresis in 1x TBE buffer at 100 V for 1.75 h. Gels were stained using SYBR Green I Nucleic Acid Gel Stain (Thermo) for 30 min. After five washes with 20 mL 1x TBE, gels were imaged using an ImageQuant LAS 4000 imager under the SYBR Green setting.

## Bioluminescence assays

Cells from overnight cultures of *V. harveyi* were pelleted by centrifugation at 21,100 x *g* (Eppendorf 5424) and resuspended in fresh LM medium. Flasks containing 25 mL of LM medium were inoculated with the washed cells, normalizing each culture to a starting $OD_{600}$ = 0.005. Culture flasks were incubated with shaking at 30˚C. Every 45 min, bioluminescence and $OD_{600}$ were measured using a Tri-Carb 2810 TR scintillation counter and DU800 spectrophotometer, respectively. *A. fischeri* cultures were grown as described for *V. harveyi* and bioluminescence was measured using a Tri-Carb 2810 TR scintillation counter when the $OD_{600}$ = 1. To assay regulation of *luxCDABE* by LuxT, *E. coli* BW25113

harboring three plasmids, described in the legend to S6 Fig, was grown in LB medium for 16 h at 30˚C. Cells from cultures were pelleted by centrifugation at 21,100 x $g$ (Eppendorf 5424) and resuspended in PBS. Bioluminescence and $OD_{600}$ were measured as above. RNA was harvested as described below for qRT-PCR analysis of *luxT* overexpression. Transcriptional output from *VIBHAR_RS11785*, *VIBHAR_RS11620*, *VIBHAR_RS16980*, and *VIBHAR_RS25670 lux* fusions was measured from *V. harveyi* strains grown to $OD_{600}$ = 1 in LM medium using a Tri-Carb 2810 TR scintillation counter. $P_{qrr1}$-*luxCDABE* activity was measured in *V. cholerae* strains using a BioTek Synergy Neo2 Multi-Mode Reader (BioTek, Winooski, VT, USA).

## Quantitative real-time PCR analyses

Cells from overnight cultures of *V. harveyi* or *A. fischeri* were pelleted by centrifugation at 21,100 x $g$ (Eppendorf 5424) and the cells were resuspended in fresh LM medium. 25 mL LM medium was inoculated with the washed cells, normalizing each culture to a starting $OD_{600}$ = 0.005. The cultures were grown shaking at 30˚C. At the desired cell densities, RNA was harvested from three independent cultures using the RNeasy mini kit (Qiagen #74106). RNA levels were normalized to 200 ng/μL and the samples were treated in two sequential reactions with DNase (Turbo DNA-free Kit, Thermo Fisher AM1907). cDNA was generated from 1 μg of RNA using Superscript III Reverse Transcriptase (Thermo Fisher, 18080093) as previously described [19]. Real-time PCR was performed using a QuantStudio 6 Flex Real-Time PCR detection system (Thermo Fisher) and PerfeCTa SYBR Green FastMix (Quantabio, 95074) as previously described [19]. In every case, 10 μL reactions were analyzed in quadruplicate technical replicates. Control reactions were performed with samples lacking reverse transcriptase and with samples lacking cDNA templates. Relative transcript levels were measured and normalized to an internal *hfq* control gene using a comparative $\Delta\Delta C_T$ method. qRT-PCR primers are listed in S2 Table.

## Western blot analyses

Overnight cultures of WT and Δ*luxT V. harveyi* strains harboring either *aphA-3xFLAG* or *3xFLAG-luxR* at their native loci were pelleted by centrifugation at 21,100 x $g$ (Eppendorf 5424) and resuspended in fresh LM medium. Flasks containing 125 mL LM medium were inoculated with the washed cells, normalizing the starting $OD_{600}$ of each culture to 0.00001. When the cultures reached the desired cell densities, cells equivalent to 1 $OD_{600}$ were pelleted by centrifugation at 2,808 x $g$ for 10 min (Eppendorf 5810 R) and the pellets were flash frozen. Next, cells were lysed by resuspension in 150 μL of buffer containing 1x BugBuster (Sigma), 1x Halt Protease Inhibitors (Thermo Fisher), 0.5% Triton X-100 (Sigma), and 50 μg/mL lysozyme (Sigma). After incubation at room temperature for 30 min, proteins were solubilized in 1x SDS-PAGE buffer for 1 h at 37˚C. Samples were loaded onto 4–20% TGX Stain-Free gels (Bio-Rad, #17000435) and subjected to electrophoresis at 50 mA for 30 min. Total loaded protein in the Stain-Free gel was visualized using an ImageQuant LAS 4000 imager using the EtBr setting. A second Stain-free gel was used for Western blot and was loaded with total protein levels normalized according to band intensities on the first gel. The normalization was verified by imaging. A dominant band from this gel image serves as a loading control in Fig 2E. FLAG-tagged protein detection was performed as previously reported [70] using an Anti-FLAG M2-Peroxidase (HRP) antibody (Sigma, A8592) and bands were visualized using an ImageQuant LAS 4000 imager.

## Fluorescence reporter assays

Fluorescent reporter plasmids are listed in S3 Table. The primers used to construct them are listed in S2 Table. Cells in overnight cultures of *Vibrio* or *Aliivibrio* strains harboring transcriptional or translational fluorescent reporter plasmids were pelleted by centrifugation at 21,100 x *g* (Eppendorf 5424) and washed in AB medium. AB medium was inoculated with the washed cells, normalizing each to $OD_{600}$ = 0.005. 150 μL of the cultures were transferred to clear-bottom 96-well plates (Corning) in quadruplicate technical replicates. 50 μL of mineral oil was added to each well to prevent evaporation. The plates were shaken at 30°C, and fluorescence and $OD_{600}$ were monitored over a 24 h period using a BioTek Synergy Neo2 Multi-Mode Reader. Relative fluorescence values represent the values when the $OD_{600}$ reached 0.3 or 0.6, as indicated in the figure legends, for each sample. The $OD_{600}$ values are the cell densities at which maximal differences between experimental and control reporter outputs could be measured.

## Hemolysis assays

Cells in overnight cultures of *V. harveyi* were pelleted by centrifugation at 21,100 x *g* (Eppendorf 5424) and resuspended in fresh LM medium. Culture densities were normalized to $OD_{600}$ = 1, and 2 μL of each culture were spotted onto a TSA plate containing 5% sheep's blood (Thermo Fisher, R060312). The plates were incubated at 30°C for 72 h and imaged above a white light. To measure hemolysis activity in liquid cultures, *V. harveyi* strains were grown for 24 h in AB medium. Cells were pelleted by centrifugation at 21,100 x *g* (Eppendorf 5424), and the clarified culture fluids were filtered through 0.22 μm filters (Sigma, SLGP033RB). Hemolysis of defibrinated sheep's blood cells (Thomas Scientific, DSB030) was measured as previously described [71, 72]. Briefly, mixtures containing 1% blood cells in PBS and 25% of the filtered fluids were incubated for 2 h at 37°C in in a 96-well plate. 1% blood cells were incubated in $ddH_2O$ or PBS as the positive and negative control, respectively. Following incubation, the plate was subjected to centrifugation at 1,000 x *g* (Eppendorf 5810 R) for 5 min at 4°C, and 100 μL of the resulting supernatants were transferred to a clean 96-well plate. Absorbance at 415 nm, indicative of blood cell lysis, was measured using a BioTek Synergy Neo2 Multi-Mode Reader.

## Bioinformatic analyses

Genomic DNA sequences of 418 *Vibrionaceae* family members were downloaded from the GenBank database (ftp.ncbi.nlm.nih.gov/genomes/genbank/bacteria/) [73]. To identify genes encoding *qrr* or *luxT*, the chromosomes were scanned for regions similar to the template sequences of *qrr* or *luxT*. As the query for *qrr* genes, we used the 3'-most 31 nucleotides of *V. harveyi qrr*1, which are highly homologous among all the *qrr* genes in *V. harveyi*, *V. cholerae*, *A. fischeri*, and *V. parahaemolyticus* [19, 20, 26]. The DNA encoding the entire *V. harveyi luxT* gene was used as the probe to identify other *luxT* genes. Local sequence alignments were performed in MATLAB (Mathworks, 2020) using the Smith-Waterman (SW) algorithm [74]. The standard scoring matrix NUC44 (see ftp.ncbi.nih.gov/blast/matrices/) was used to compute similarity scores, which take into account both the length and sequence similarity of the alignment. Cutoff values for the similarity scores yielded from the SW algorithm were set to 30 for *qrr* genes and 100 for *luxT* genes. Genes identified as possible *luxT* homologs were verified to encode TetR family transcriptional regulators. Species lacking either *qrr*1 or *luxT* were excluded from further phylogenetic analyses.

Multiple sequence alignments were performed using T-Coffee [75]. Phylogenetic analyses and tree building were performed in MATLAB. To construct the phylogenetic tree based on

the putative LuxT binding regions residing upstream of *qrr*1 genes (see Fig 7), using the maximum-likelihood based Jukes-Cantor model [76], we first computed the pairwise difference scores between the 30 nucleotides upstream of the $\sigma^{54}$ binding sites in the *qrr*1 promoter regions for every two species. The unweighted pair group method with arithmetic mean (UPGMA) was subsequently used to progressively build a hierarchy of species clusters [77]. In brief, each species was initially represented by one node. At each clustering step, the pair of nodes with the minimal difference score were clustered into a new node. The arithmetic means of the difference scores between this node pair and each of the other nodes were then assigned to be the difference scores between the newly clustered node and other nodes. The sequence logos were generated by WebLogo [78, 79].

## Statistical methods

All statistical analyses were performed using GraphPad Prism software. Error bars correspond to standard deviations of the means of three biological replicates.

## Supporting information

**S1 Table. Strains used in this study.**
(PDF)

**S2 Table. Oligonucleotides used in this study.**
(PDF)

**S3 Table. Plasmids used in this study.**
(PDF)

**S1 Fig. LuxT binds upstream of *luxO*.** EMSA showing binding of LuxT-6xHis to 95 bp DNA fragments containing the WT *luxO* promoter (left) and the *luxO* promoter in which the 50 nucleotides previously shown to be crucial for LuxT binding were randomized (right). DNA and protein concentrations as in Fig 2A.
(PDF)

**S2 Fig. The *luxO*-*qrr*1 locus.** The *V. harveyi* genomic DNA region harboring the LuxO-P, LuxT, and $\sigma^{54}$ binding sites. The sites are labeled in relation to the *qrr*1 +1 transcriptional start site, which is also designated. Colors as in Fig 3A.
(PDF)

**S3 Fig. LuxT does not repress *qrr*2-5. (A)** Relative fluorescence values (mRuby3/$OD_{600}$) of the indicated *V. harveyi* strains harboring a plasmid-borne $P_{qrr2}$-*mRuby3* transcriptional reporter. Values represent relative fluorescence at $OD_{600}$ = 0.6 for each sample. **(B-D)** As in A, except the strains harbor $P_{qrr3}$-*mRuby3*, $P_{qrr4}$-*mRuby3*, and $P_{qrr5}$-*mRuby3*, respectively. In all panels, error bars represent standard deviations of the means of $n$ = 3 biological replicates. Unpaired two-tailed $t$ tests with Welch's correction were performed comparing the indicated two samples. $p$-values: ns $\geq$ 0.05.
(PDF)

**S4 Fig. LuxT activates *luxCDABE* independently of Qrr1.** Density-dependent bioluminescence production from WT (black), Δ*luxT* (blue), Δ*qrr*1 (green), and Δ*qrr*1 Δ*luxT* (orange) *V. harveyi* strains. Relative light units (RLU) are counts/min mL$^{-1}$ per $OD_{600}$. Error bars represent standard deviations of the means of $n$ = 3 biological replicates.
(PDF)

**S5 Fig. LuxT does not bind the *luxCDABE* promoter. (A)** Diagram of the *luxCDABE* promoter region. Black striped boxes represent known LuxR binding sites [34]. The black lines labeled 1 to 6 show the ~100 bp overlapping DNA fragments that were amplified and used as probes. The probes span the region -405 to +81 relative to the *luxC* start codon. **(B-D)** EMSAs measuring LuxT-6xHis binding to Probes 1–6 from panel A. DNA and protein concentrations as in Fig 2A.
(PDF)

**S6 Fig. LuxT does not directly activate *luxCDABE* in *E. coli*. (A)** Bioluminescence production from *E. coli* BW25113 harboring *luxCDABE* expressed from its native promoter on a plasmid (pBB1). The *E. coli* carries two additional plasmids, as indicated. - denotes the empty parent vector. + denotes the p*luxR* and/or the p*luxT* plasmid, encoding IPTG inducible *luxR* and arabinose inducible *luxT*, respectively. Strains were grown for 16 h in LB containing 0.5 mM IPTG in the absence (black) or presence (gray) of 0.2% arabinose. Relative light units (RLU) are counts/min mL$^{-1}$ per OD$_{600}$. **(B)** qRT-PCR measurements of *luxT* transcript levels in the *E. coli* strains harboring the p*luxT* plasmid from panel A. **(C)** Cell densities (OD$_{600}$) of the strains in panel A after 24 h of growth. For panels B and C, the labeling and color schemes are as in panel A. In all panels, error bars represent standard deviations of the means of *n* = 3 biological replicates.
(PDF)

**S7 Fig. LuxT activation of *luxCDABE* does not depend on known QS genes. (A-D)** Density-dependent bioluminescence production from the designated *V. harveyi* strains that possess (black) and lack (blue) *luxT*. Relative light units (RLU) are counts/min mL$^{-1}$ per OD$_{600}$. Error bars represent standard deviations of the means of *n* = 3 biological replicates.
(PDF)

**S8 Fig. LuxT does not regulate *luxR* and *aphA*.** qRT-PCR measurements of *luxR* and *aphA* transcript levels in WT (black) and Δ*luxT* (blue) *V. harveyi* at LCD (OD$_{600}$ = 0.05). Error bars represent standard deviations of the means of *n* = 3 biological replicates. Unpaired two-tailed *t* tests with Welch's correction were performed comparing WT to Δ*luxT*. *p*-values: ns ≥ 0.05.
(PDF)

**S9 Fig. LuxT activates Qrr target mRNAs independently of Qrr1. (A)** Transcript levels of the indicated *VIBHAR_RS* genes as measured by qRT-PCR in *V. harveyi luxO* D61E and *V. harveyi luxO* D61E Δ*luxT* strains at OD$_{600}$ = 1. Unpaired two-tailed *t* tests with Welch's correction were performed comparing *V. harveyi luxO* D61E to *V. harveyi luxO* D61E Δ*luxT*. *p*-values: ns ≥ 0.05, ** < 0.01, *** < 0.001, **** < 0.0001. **(B)** qRT-PCR measurements of transcript levels of the indicated *VIBHAR_RS* genes in the designated *V. harveyi* strains at OD$_{600}$ = 1. Different letters indicate significant differences between strains, *p* < 0.05 (two-way analysis of variation (ANOVA) followed by Tukey's multiple comparisons test). In both panels, error bars represent standard deviations of the means of *n* = 3 biological replicates.
(PDF)

**S10 Fig. LuxT activates the transcription of target genes.** Activities of *lux* transcriptional fusions to the indicated promoters were measured in the designated *V. harveyi* strains at OD$_{600}$ = 1. Relative light units (RLU) are counts/min mL$^{-1}$ per OD$_{600}$. Error bars represent standard deviations of *n* = 3 biological replicates. Unpaired two-tailed *t* tests with Welch's correction were performed comparing *V. harveyi luxA*::Tn*5* to *V. harveyi luxA*::Tn*5* Δ*luxT*. *p*-values: ** < 0.01, **** < 0.0001.
(PDF)

**S11 Fig. Qrr1 overexpression represses translational reporter constructs.** Relative fluorescence (mVenus/OD$_{600}$) of WT *V. harveyi* harboring a plasmid encoding a translational reporter to the indicated *VIBHAR_RS* gene transcribed from the aTc inducible *tetA* promoter. The *V. harveyi* strains also carry IPTG-inducible *qrr*1 on a plasmid (p*qrr*1) or the empty parent vector (pControl). All strains were grown in the presence of 0.5 mM IPTG. Strains were grown in the absence and presence of 100 ng mL$^{-1}$ aTc (- aTc and + aTc, respectively). Values represent relative fluorescence at OD$_{600}$ = 0.3 for each sample. Error bars represent standard deviations of the means of $n$ = 3 biological replicates. Different letters indicate significant differences between strains, $p < 0.05$ (two-way analysis of variation (ANOVA) followed by Tukey's multiple comparisons test).
(PDF)

**S12 Fig. LuxT and Qrr1 control aerolysin production.** Hemolytic activity present in the indicated *V. harveyi* cell-free culture fluids as judged by lysis of defibrinated sheep's blood. Culture fluids were collected after 24 h of growth in AB medium. Hemolytic activity was normalized to the activity of ddH$_2$O [A$_{415}$(sample)/A$_{415}$(ddH$_2$O) x 100]. Error bars represent standard deviations of the means of $n$ = 3 biological replicates. Unpaired two-tailed $t$ tests with Welch's correction were performed comparing two samples, as indicated. $p$-values: *** $< 0.001$, **** $< 0.0001$.
(PDF)

**S13 Fig. LuxT does not appear to control *qrr*1 in *V. cholerae* or *V. parahaemolyticus*. (A)** Activity of a *V. cholerae* P$_{qrr1}$-*luxCDABE* transcriptional reporter in the indicated *V. cholerae* strains. **(B)** Relative fluorescence of a *V. parahaemolyticus* P$_{qrr1}$-*mRuby3* transcriptional reporter measured in the indicated *V. parahaemolyticus* strains. Relative light production (panel A) and relative fluorescence (panel B) represent values when OD$_{600}$ = 0.6 for each sample. Error bars represent standard deviations of the means of $n$ = 3 biological replicates. Unpaired two-tailed $t$ tests with Welch's correction were performed comparing two samples, as indicated. $p$-values: ns $\geq 0.05$.
(PDF)

**S1 Data. Numerical data for Figs 2B, 2C, 2D, 3B, 3C, 3D, 4A, 5A, 5B, 5C, 5D, 6B, 6C, 6D, 7A and 7B.**
(XLSX)

**S2 Data. Numerical data for S3A, S3B, S3C, S3D, S4, S6A, S6B, S6C, S7A, S7B, S7C, S7D, S8, S9A, S9B, S10, S11, S12, S13A and S13B Figs.**
(XLSX)

## Acknowledgments

We thank members of the Bassler laboratory and Dr. Ned Wingreen for insightful discussions and suggestions.

## Author Contributions

**Conceptualization:** Michaela J. Eickhoff, Bonnie L. Bassler.

**Data curation:** Michaela J. Eickhoff, Bonnie L. Bassler.

**Formal analysis:** Michaela J. Eickhoff, Chenyi Fei, Bonnie L. Bassler.

**Funding acquisition:** Bonnie L. Bassler.

**Investigation:** Michaela J. Eickhoff, Chenyi Fei, Xiuliang Huang.

**Methodology:** Michaela J. Eickhoff, Bonnie L. Bassler.

**Project administration:** Bonnie L. Bassler.

**Resources:** Michaela J. Eickhoff, Chenyi Fei, Xiuliang Huang, Bonnie L. Bassler.

**Software:** Chenyi Fei.

**Supervision:** Bonnie L. Bassler.

**Validation:** Michaela J. Eickhoff, Bonnie L. Bassler.

**Visualization:** Michaela J. Eickhoff, Chenyi Fei, Bonnie L. Bassler.

**Writing – original draft:** Michaela J. Eickhoff, Chenyi Fei, Xiuliang Huang, Bonnie L. Bassler.

**Writing – review & editing:** Michaela J. Eickhoff, Bonnie L. Bassler.

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
