## [Decision Letter · Decision Letter 0]

22 Feb 2021

**Subject: Minor revision required**

Dear Bonnie, dear co-authors,

Thank you very much for submitting your Research Article entitled 'LuxT controls specific quorum-sensing-regulated behaviors in Vibrionaceae spp. via repression of qrr1, encoding a small regulatory RNA' to PLOS Genetics.

The manuscript was now fully evaluated at the editorial level and by two independent expert peer reviewers. The reviewers appreciated the attention to an important topic but nonetheless identified some minor concerns that we ask you address in a revised manuscript. 

We therefore ask you to modify the manuscript according to the reviewers' recommendations. Your revisions should address the specific points made by each reviewer. Please take reviewer #1's comments about the transcriptional activation into special consideration by providing additional experimental data supporting this claim. Alternatively, you could revise the text by discussing alternative options, as suggested by the reviewer.

You can use this link to log into the system when you are ready to submit a revised version, having first consulted our Submission Checklist.

[LINK]

Please let us know if you have any questions while making these revisions. And we kindly ask you to accept our apologies for the slight delay in getting the manuscript back to you. As you can imagine, the current COVID situation makes every step of the peer-reviewing process more complicated and time consuming due to the preoccupation of the people involved. 

Best wishes,

Melanie

Melanie Blokesch

Associate Editor

PLOS Genetics

Lotte Søgaard-Andersen

Section Editor: Prokaryotic Genetics

PLOS Genetics

Reviewer's Responses to Questions

**Comments to the Authors:**

Reviewer #1: Summary:

In this manuscript submitted to PLOS Genetics, Eickhoff and colleagues describe the molecular mechanism by which the transcription factor LuxT regulates the expression of quorum sensing regulated genes via the Qrr1 sRNA in Vibrio harveyi. The five Qrr sRNAs were previously shown to regulate gene expression in this organism by base-pairing with complementary sequences in target mRNAs altering their stability or translation efficiency. Most, but not all of the target mRNAs identified so far are similarly regulated by all five Qrr sRNAs. In this manuscript under review, the authors provide evidence that the expression of Qrr1 sRNA is regulated differently than the other four Qrr sRNAs. Specifically, the authors show that LuxT represses transcription of qrr1 (Fig. 2) and that LuxT reduces the levels of Qrr1 sRNA, but not Qrr2, Qrr3, Qrr4, or Qrr5 (Fig. 3). The authors then provide evidence that LuxT increases expression of the luciferase genes by a mechanism that is independent of Qrr1 (Figs 2 and S4) and activates expression of 9 other genes by a mechanism that is dependent (Fig. 5 and S10) and independent (Fig. 4 and S9) of Qrr1. The authors then assess the conservation of transcriptional repression of qrr1 by LuxT homologs among several related species (Vibrio cholerase, Vibrio parahaemolyticus, Aliivibrio fischeri; Figs 6, 7, S12).

Overall, this manuscript is well written and well thought out and clearly demonstrates that Qrr1 is regulated by a mechanism distinct from the other Qrr sRNAs, i.e., via LuxT repression, and that LuxT regulates many quorum sensing-associated genes via a Qrr1 dependent and independent mechanism. Furthermore, this manuscript was a pleasure to read. A major sticking point for me was a central tenet put forth by the authors that LuxT activates the transcription of luxCDABE, VIBHAR_RS11785, VIBHAR_RS11620, VIBHAR_ RS16908, and VIBHAR_RS25670. While the authors show that the levels of these transcripts do increase in the absence of LuxT by a mechanism that is independent of the Qrr sRNAs, this increase could be due to LuxT indirectly regulating the stability of these transcripts, e.g., through another sRNA. In my opinion, the authors could resolve this conundrum by revising the text to make clear that LuxT activation of the transcription of these genes is only one possibility and to present alternative explanations supported by their results (minor revision) or to clearly show that LuxT activates the transcription of these genes using transcriptional fusions to their promoters (major revision). Other minor issues are also described below.

Major criticisms:

1. The authors indicate several times that LuxT activates the transcription of genes including VIBHAR_RS11785, VIBHAR_RS11620, VIBHAR_ RS16908, and VIBHAR_RS25670, but this conclusion is not necessarily supported by the data. It is true that the levels of these transcripts go down in a luxT deletion strain regardless of whether qrr1 is present. However, the authors' experiments do not rule out the possibility that LuxT could cause an increase in the level of these transcripts by increasing the stability of these mRNAs, e.g., through some other sRNA. To decisively demonstrate that LuxT activates the transcription of these genes either directly or indirectly, the authors could test the impact of the luxT deletion on transcriptional reporter fusions to the promoters of these genes in the presence or absence of qrr1.

Minor points:

2. Lines 394-396. The authors state based on Fig. 5E that deletion of qrr1 in the luxO D61E background strain caused increased hemolytic activity, but that result is not consistent with the quantitative assay of hemolysis (S11). The authors gloss over this fact.

3. Lines 51, 59, 69. The authors reiterate several times that the five Qrr sRNAs are redundant; however, the authors already know from their previous work (PMIDs 22229925, 23838640) that this statement is not entirely accurate.

4. It would be useful to differentiate the section titles from the body of the text in the results section by putting them in bold face type.

5. Fig. 2B. Error bars appear to be missing. If error bars are so small that they are not visible, the authors should specifically denote that.

6. Figs. 3. It would be useful if the bars in B and C were colored using the same scheme as in D.

7. Line 205. The authors conclusion that "LuxT is a LCD transcriptional activator of luxCDABE" is not necessarily supported by their data. LuxT could be affecting transcript levels by impacting mRNA stability rather than transcription. Instead, the authors could state that "LuxT activates luxCDABE expression"

8. Line 263. Change "LuxT activates luxCDABE transcription" to "LuxT activates luxCDABE expression". See prior point.

9. Line 346. Replace "transcription" with "expression". No clear data supporting an impact on transcription versus mRNA stability.

10. Lines 398 to 399, 536, 543. Present data does not necessarily support transcriptional regulation.

11. Lines 412 to 413. Take out parentheses.

12. Lines 553 to 556. Diversification of DNA sequence of LuxT binding regions does not necessarily signify differences in LuxT-mediated repression of qrr1 expression in different species. As the authors indicate earlier, LuxT specificity could co-evolve with its binding site upstream of qrr1.

13. Line 612. Isopropyl should not be capitalized.

14. Lines 691 to 704. The authors should indicate use of NTC and NRT controls.

Reviewer #2: This manuscript by Eickhoff, et al., investigates regulation of quorum sensing (QS) in Vibrio harveyi via transcriptional and post-transcriptional mechanisms. They identify LuxT (a TetR-family transcription factor) as a direct repressor of qrr1, encoding one of the five QS small RNAs in V. harveyi. There are several major findings in this study. First, LuxT appears to be a global regulator of QS and exerts control over this system via direct transcriptional control of Qrr1 production and indirect control of other QS genes, including the luciferase operon. Another layer of LuxT activity is transcriptional control of (at least a subset of) Qrr1 target genes, though it is unclear whether this is direct or indirect. The authors use a very nice set of genetic experiments to convincingly demonstrate that LuxT regulates Qrr1 target genes both transcriptionally (Qrr1-independent) and post-transcriptionally (Qrr1-dependent). They also show that LuxT is conserved among the Vibrionaceae but that LuxT-dependent repression of qrr1 is only conserved in a subset of species. The findings add an additional layer of complexity to Vibrio QS systems and suggest a mechanism by which different signals could feed into the system to independently control activities of individual Qrr sRNAs. The study is of high quality. The experiments have the appropriate controls and the logic and interpretation of results is easy to follow. While I’m convinced that LuxT has the capacity to control QS genes both transcriptionally and post-transcriptionally, the post-transcriptional Qrr1-dependent effects could only be observed in reporter strains where the transcriptional effects were negated. Thus, the physiological relevance of the dual-control remains a mystery because the transcriptional effects of LuxT are clearly dominant. Moreover, qrr1 is the only gene that could be convincingly demonstrated as a direct transcriptional target of LuxT, so the mechanism of the other transcriptional effects remains unclear. Nevertheless, this work provides a new and elegant example of signal integration into a complex regulatory system and interesting multi-level control of gene expression.

Other comments:

• I’m not convinced that the data support the assertion in the title. The LuxT regulation of QS behaviors was mostly via the transcriptional, Qrr1-independent pathway.

• Line 130-131: What is meant by “distinct production profiles”? The list with “>” symbols suggests overall abundance. Is it also timing?

• Lines 159-163: When the Qrr1 targets are introduced it would be helpful to state whether they are exclusively regulated by Qrr1 or if they are also regulated by other Qrr sRNAs, and provide the reference for the paper that identified them as Qrr1 targets.

• It would be very helpful to provide the legends indicating what bar colors represent for all figures, similar to Fig. S9B. It allows the meaning of the data to be more immediately obvious without having to search for the figure legends.

• In Fig. S9, most of the targets do not appear to be regulated by Qrr1 at all – i.e., there is no difference between isogenic qrr1+ and delta qrr1 strains. I do understand the model in Fig. 4B that attempts to explain why a Qrr1 effect might be less apparent in the context of the transcriptional control of LuxT, but that implies that the LuxT effect on transcription is all or nothing. In other words, in the presence of LuxT, transcription would have to be so high that post-transcriptional “activation” (repressing the Qrr1 repressor) would be irrelevant, and in the absence of LuxT, transcripts would have to be at such low levels that further repression by Qrr1 could not be measured. Is there any evidence for this? It looks like the different targets are present at a range of concentrations (though I might be misjudging because the data are reported as relative transcript levels). Addressing this would go a long way toward addressing my primary concern about this manuscript.

• Related to comment above: when Qrr1 targets are put under control of a heterologous promoter (Fig. 5), the Qrr1 effect is immediately evident, and the fold change between qrr1+ and deltaqrr1 strains appears fairly substantial for at least a couple of the targets. This again makes me think we should be able to see at least modest Qrr1-dependent regulation for some of these targets in Fig S9B.

• Fig. 5E hints at a Qrr1-dependent increase in aerolysin production, but the difference between luxO D61E and luxO D61E delta qrr1 is not evident in the quantitated hemolysin Fig. S11. This should be clarified.

**Have all data underlying the figures and results presented in the manuscript been provided?**

Reviewer #1: Yes

Reviewer #2: Yes

PLOS authors have the option to publish the peer review history of their article (what does this mean?). If published, this will include your full peer review and any attached files.

Reviewer #1: No

Reviewer #2: No

---

## [Editor Report · Decision Letter 1]

20 Mar 2021

Dear Dr Bassler, dear Bonnie,

We are pleased to inform you that your manuscript entitled "LuxT controls specific quorum-sensing-regulated behaviors in Vibrionaceae spp. via repression of qrr1, encoding a small regulatory RNA" has been editorially accepted for publication in PLOS Genetics. Congratulations!

Yours sincerely,

Melanie

Melanie Blokesch

Associate Editor

PLOS Genetics

Lotte Søgaard-Andersen

Section Editor: Prokaryotic Genetics

PLOS Genetics

Comments from the reviewers (if applicable):

**Data Deposition**

http://datadryad.org/submit?journalID=pgenetics&manu=PGENETICS-D-21-00004R1

**Press Queries**

---

## [Editor Report · Acceptance letter]

29 Mar 2021

PGENETICS-D-21-00004R1 

LuxT controls specific quorum-sensing-regulated behaviors in *Vibrionaceae* spp. via repression of *qrr*1, encoding a small regulatory RNA 

Dear Dr Bassler, 

We are pleased to inform you that your manuscript entitled "LuxT controls specific quorum-sensing-regulated behaviors in *Vibrionaceae* spp. via repression of *qrr*1, encoding a small regulatory RNA " has been formally accepted for publication in PLOS Genetics! Your manuscript is now with our production department and you will be notified of the publication date in due course.

With kind regards,

Alice Ellingham

PLOS Genetics

On behalf of:
